# CodeRL: Mastering Code Generation through Pretrained Models and Deep Reinforcement Learning

**Hung Le**,* **Yue Wang**,* **Akhilesh Deepak Gotmare, Silvio Savarese, Steven C.H. Hoi** †
Salesforce Research
`https://github.com/salesforce/CodeRL`

## Abstract

Program synthesis or code generation aims to generate a program that satisfies a problem specification. Recent approaches using large-scale pretrained language models (LMs) have shown promising results, yet they have some critical limitations. In particular, they often follow a standard supervised fine-tuning procedure to train a code generation model from natural language problem descriptions and ground-truth programs only. Such paradigm largely ignores some important but potentially useful signals in the problem specification such as unit tests, which thus results in poor performance when solving complex unseen coding tasks. To address the limitations, we propose "CodeRL", a new framework for program synthesis tasks through pretrained LMs and deep reinforcement learning (RL). Specifically, during training, we treat the code-generating LM as an actor network, and introduce a critic network that is trained to predict the functional correctness of generated programs and provide dense feedback signals to the actor. During inference, we introduce a new generation procedure with a critical sampling strategy that allows a model to automatically regenerate programs based on feedback from example unit tests and critic scores. For the model backbones, we extended the encoder-decoder architecture of CodeT5 with enhanced learning objectives, larger model sizes and better pretraining data. Our method not only achieves new SOTA results on the challenging APPS benchmark, but also shows strong zero-shot transfer capability with new SOTA results on the simpler MBPP benchmark.

## 1  Introduction

Considering program synthesis as a sequence-to-sequence task, pretrained language models (LMs) [Hendrycks et al., 2021, Chen et al., 2021a, Austin et al., 2021] can be adapted to receive input sequence as problem specification in natural language and generate a sequence of codes as the output program (see Figure 1, right, for an example). While these models achieve promising results, especially in basic programming tasks [Chen et al., 2021a, Austin et al., 2021], we observe that they still fail to generate codes to solve complex problems [Hendrycks et al., 2021, Li et al., 2022].

There are two main limitations. First, current models are trained using a conventional next-token prediction (NTP) objective which maximizes the next ground-truth token likelihood. As noted in NLP domains [Bengio et al., 2015, Ranzato et al., 2016], training models only with next-token prediction objective in a "teacher-forcing" manner often leads to accumulating errors during test time when tokens are generated by conditioning on previously sampled tokens, not the ground-truth tokens. This issue becomes more serious in the domain of program synthesis, where token-matching scores such as BLEU [Papineni et al., 2002, Ren et al., 2020] are more appropriate in partial program synthesis tasks (i.e. code completion) [Husain et al., 2019] but have failed to measure the functional correctness

---

*Equal contribution.
†Corresponding authors: {hungle, shoi}@salesforce.com

36th Conference on Neural Information Processing Systems (NeurIPS 2022).

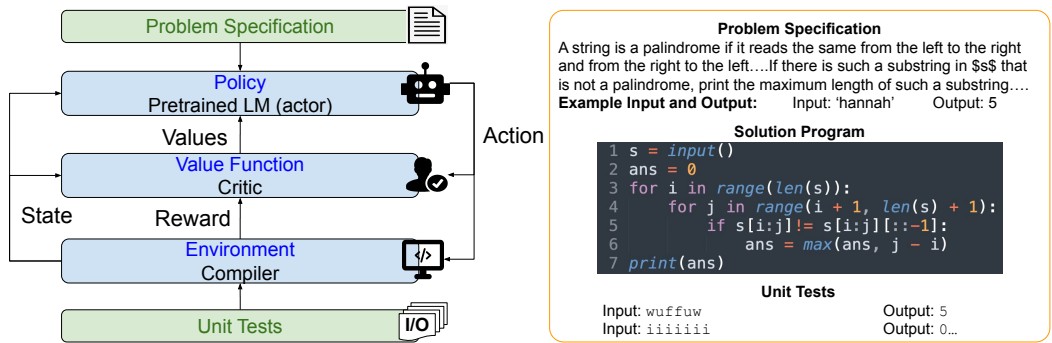

Figure 1: **An example program synthesis task (Right):** Each task is defined by a problem specification in natural language, often containing example input and output pairs. The expected output is a program to be checked for functional correctness against some unit tests. **A high-level overview of our CodeRL framework for program synthesis (Left):** we treat a pretrained code language model (LM) as a stochastic policy, code generations as actions, and rewards can be estimated based on the unit test results of output programs from the compiler (environment).

of complete programs [Hendrycks et al., 2021, Chen et al., 2021a]. Training only with NTP objective is hence, not ideal to tackle full program generation to solve programming problems.

Secondly, current models fail to utilize the potential meaningful signals from unit tests, which directly determine the model performance by the functional correctness of programs. Current approaches neglect this important signal during model optimization as well as generation procedure. During optimization, unit tests could be factored into learning objectives to match the final goal of generating semantically correct programs. During inference, since unit tests are often parts of problem description (i.e. example unit tests), they are potentially powerful to further improve output programs.

To address the above issues, we introduce "CodeRL", a new framework to improve pretrained LMs for program synthesis tasks through reinforcement learning (see Figure 1, left). Specifically, we propose a training strategy that optimizes pretrained LMs for program synthesis tasks in an actor-critic approach [Konda and Tsitsiklis, 1999, Sutton et al., 1999]. We treat the pretrained LM as an actor network and synthetically sample sequences from this actor, including both correct and incorrect programs. These program samples are passed to a critic model which is trained as an error predictor to assess the functional correctness of these samples. We use the token-level hidden states extracted from the learned critic model to estimate the values/scores of output tokens of these synthetic samples. The actor network is then finetuned on these synthetic samples weighted by their critic scores. During inference, we introduce a new generation procedure that involves example unit tests and a critic model to filter and select sub-sequences. These sub-sequences are utilized as seeds that condition the model to resample new tokens and obtain new output programs. This approach allows the model to automatically refine output programs based on their functional correctness during test time.

We extend CodeT5 with better pretraining strategies as the foundation model for CodeRL. Our comprehensive experiments show that our models can achieve SOTA performance on the challenging APPS benchmark [Hendrycks et al., 2021]. Specifically, our models reach more than 2% *pass@1*, 6% *pass@5*, and 19% *pass@1000*. Since our RL method is model-agnostic, we apply it to various large-scale models and achieve consistent performance gains. We further test its zero-shot transfer ability on a simpler MBPP benchmark [Austin et al., 2021], where it sets a new SOTA result of 63.0% *pass@80* over a finetuned GPT-137B's 61.4%. We release the improved CodeT5-large (770M) model which outperforms many pretrained LMs of much larger sizes.

## 2 Related Work

### 2.1 Program Synthesis

Program synthesis tasks can date back as early as the early adoption of machine learning research [Waldinger and Lee, 1969, Manna and Waldinger, 1971]. Earlier tasks include problem specifications in the form of input-output (IO) examples [Summers, 1977, Gulwani et al., 2012] and synthesis

methods are limited to probabilistic approaches [Liang et al., 2010] or simple programming concepts [Joulin and Mikolov, 2015, Kurach et al., 2015]. As deep learning methods became popular, later approaches adopt neural models to induce output programs, assuming an inductive bias given a large number of program samples [Parisotto et al., 2016, Balog et al., 2016, Devlin et al., 2017]. More recently, we witnessed the emergence of program synthesis tasks in which output programs are extended to general-purpose programming languages [Yin and Neubig, 2017, Xu et al., 2018, Chen et al., 2021a] and program specifications are fully described in natural English text [Hendrycks et al., 2021, Austin et al., 2021, Poesia et al., 2022]. These extensions have encouraged a rising number of applications of pretrained language models (LMs) to program synthesis to exploit the contextual representations learned from massive data of codes and natural languages [Feng et al., 2020, Clement et al., 2020, Wang et al., 2021, Wang and Komatsuzaki, 2021, Chen et al., 2022]. Recently, Nijkamp et al. [2022] proposed a conversational program synthesis approach with large pretrained language models. Despite impressive results in basic programming problems and initial commercial deployment[3], existing models still perform poorly against complex problems such as those from programming competitions on Codeforces [Hendrycks et al., 2021, Li et al., 2022].

**Program Synthesis in Visual Context.**   Another related line of research is program synthesis in computer vision domains such as images and videos. Early papers such as [Kulkarni et al., 2015, Yang et al., 2015] introduce inverse graphics networks to infer visual properties such as pose, shape, and lighting, of visual objects. Wu et al. [2017], Liu et al. [2019], Ellis et al. [2018] study the problem of image rendering, which transforms an image to structured and compact representations, i.e. *scene programs*. Tian et al. [2019] extends the prior work to render 3D shapes from images through *shape programs*, containing features to capture geometric and structural priors. Ganin et al. [2018] introduces an RL-based approach to render realistic images through high-level graphics programs. Sun et al. [2018]introduces program synthesis from demonstration synthetic videos to summarize the behaviors of the objects in the videos.

While this line of research has remarkable impacts on applications such as image/video editing, captioning, and extrapolating, these approaches are limited to programs of domain-specific languages defined for visual objects. For instance, in [Sun et al., 2018], programming language contains basic functions for object perception, action, and control flows. In our work, we focus on program synthesis from natural language problem specifications and the output programs are in general-purpose languages such as Python. This type of programming task can range from basic programming problems to competition-level programming tasks that require a high level of problem-solving skills.

## 2.2   Reinforcement Learning for Sequence Generation

Related to the program synthesis tasks are research domains of sequence generation, in which RL approaches have demonstrated remarkable achievements. In these domains, RL approaches are used to exploit signals from non-differentiable metrics of the task at hand. Earlier work such as [Ranzato et al., 2016] adopts this strategy with REINFORCE algorithm [Williams, 1992] to directly optimize models for sequence-based test metrics such as BLEU [Papineni et al., 2002] and ROUGE [Lin, 2004] scores for translation models. In the same domain, Bahdanau et al. [2016] introduced an actor-critic framework [Sutton, 1984, Konda and Tsitsiklis, 1999]. In visual captioning domains, Rennie et al. [2017], Wang et al. [2018] proposed to use RL to optimize image captioning models using variants of CIDEr scores [Vedantam et al., 2015]. Alternatively, Ren et al. [2017] derived a new goal-oriented return estimate using visual-semantic embedding. Johnson et al. [2017], Trivedi et al. [2021] introduce program generation as an auxiliary task to learn interpretable policies in question-answering and synthetic navigation tasks.

Different from prior domains, in program synthesis, Austin et al. [2021], Chen et al. [2021a], Li et al. [2022] demonstrated very low correlation between token-based similarity metrics and functional correctness of programs. Hence, it is not trivial to define an appropriate optimization goal in this domain. We propose to exploit unit test signals, which directly exhibit the functional correctness of programs, during both - model optimization and test-time generation stages. More related to our work are RL-based program synthesis [Guu et al., 2017, Bunel et al., 2018, Liang et al., 2018, Zhong et al., 2018] and execution-guided synthesis approaches [Ellis et al., 2019, Chen et al., 2021b]. However, these are limited to programming languages defined within a specific application domain only.

---

[3]https://copilot.github.com/

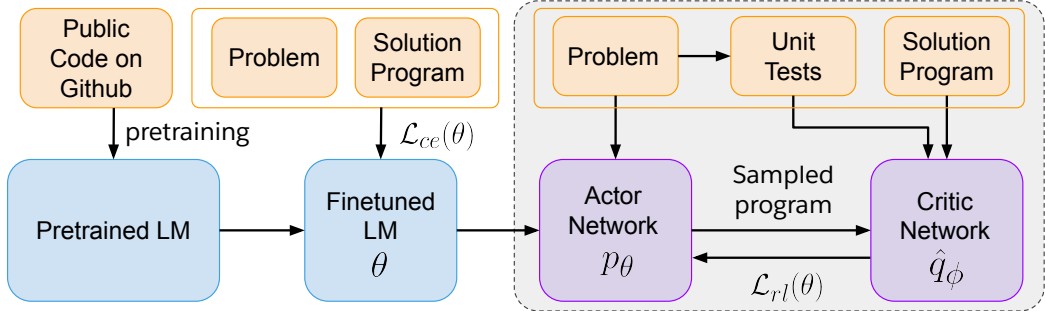

Figure 2: **Overview of our actor-critic framework to optimize pretrained LMs for program synthesis:** We treat the LM as an actor network and sample synthetic samples from this actor. Another neural network is trained as a critic model to evaluate these synthetic samples based on their probabilities of passing unit tests. The returns are estimated based on critic scores and finally factored into the learning objective $\mathcal{L}_{rl}$ to finetune the actor LM network using synthetic samples.

## 3 CodeRL

### 3.1 Program Synthesis Task

Following a sequence-to-sequence approach, the program synthesis task contains a problem description as an input sequence $D$ and an output sequence of program $\hat{W} = (\hat{w}_1, ..., \hat{w}_T), \hat{w}_t \in \mathcal{V}$ [4] that can solve the problem. The output at each decoding step $t$ is a distribution over the vocabulary $\mathcal{V}$, computed by the softmax function $\hat{w}_t \sim \text{softmax}(\text{Linear}(s_t))$ where $s_t$ is the contextual hidden state at decoding step $t$. Conventionally, during train time, model parameters, $\theta$, are learned by maximizing the likelihood of the ground-truth reference programs. Denoting $W = (w_1, ...w_T)$ as the ground-truth program, the objective is to minimize the cross-entropy loss:

$$\mathcal{L}_{ce}(\theta) = -\sum_t \log p_\theta(W|D) = -\sum_t \log[p_\theta(w_t|w_{1:t-1}, D)] \tag{1}$$

where the conditional probability $p_\theta$ is parameterized following the above softmax function. During test time, models generate sequences of programs by autoregressively sampling token $\hat{w}_t$ from the distribution $p_\theta(.|\hat{w}_{1:t-1}, D)$. Models are evaluated against unit tests corresponding to the problem. Each test includes a pair of input and ground-truth output. In real-world program synthesis tasks [Hendrycks et al., 2021], *example unit tests* are often given as parts of the problem specification.

### 3.2 Pretraining Language Models on Code

We adopt Transformer models as the backbone of our program synthesis systems. Specifically, this paper extends the CodeT5 model [Wang et al., 2021] as a foundation model for CodeRL.

**CodeT5.** CodeT5 [Wang et al., 2021] is a multi-lingual code-aware language model pretrained on large-scale source code corpora curated from Github. With a unified encoder-decoder architecture, CodeT5 achieves state-of-the-art performance in a wide range of code intelligence tasks in the CodeXGLUE benchmark [Lu et al., 2021] including both code understanding and generation tasks.

**Improving Pretraining Data.** We enlarge the Python pretraining dataset using the recently released large-scale Github Code dataset[5]. We have compiled public, non-personal information from GitHub consisting of permissively licensed Python code ("mit", "apache-2", "bsd-3-clause", "bsd-2- 126 clause", "cc0-1.0", "unlicense", "isc"). The resulting Python dataset (GCPY) has 10.5B tokens and is 10x larger than the CodeSearchNet (CSN) corpus [Husain et al., 2019] used in the original CodeT5 [Wang et al., 2021].

---

[4]For simplicity, we use $T$ as the notation of sequence length for all sequences which can actually be variable.
[5]`https://huggingface.co/datasets/lvwerra/github-code`

**Improving Pretraining Objective.** While pretraining tasks in CodeT5 like masked span prediction (MSP) benefit code understanding tasks, they have a large discrepancy with program synthesis objectives. To mitigate this gap, we introduce a pretraining task of next-token prediction (NTP) into CodeT5. Specifically, we uniformly sample a pivot location for each code sample, then pass the content preceding the pivot to the encoder and the remaining to the decoder. To control the length of input and output sequences, we restrict the pivot within 10% to 90% of the original sequence.

### 3.3 Program Synthesis as an RL Problem

We propose to formulate the Program Synthesis as an RL problem and apply an actor-critic RL approach to improve the performance of a pretrained LM by exploiting the unit test signals in both model optimization (see Figure 2) and generation procedure (see Figure 3).

More formally, we can view the learned parameters of an LM model, $\theta$ as a stochastic *policy*, which decides an *action* as the prediction of each token. Following each action, an LM model updates its hidden state representations which are used by the policy to determine the next action in the next decoding step. At the end of the generation episode (i.e. an *<endoftext>* token is observed), the LM model receives a *return* $r$ measured by the functional correctness of the generated program. The goal of RL finetuning is to minimize the expected return:

$$\mathcal{L}_{rl}(\theta) = -\mathbb{E}_{W^s \sim p_\theta}[r(W^s)] \tag{2}$$

where $W^s = (w_1^s, ..., w_T^s)$ is a synthetic sample in which each token $w_t^s$ is sampled by the LM model at decoding time step $t$. Following the REINFORCE algorithm [Williams, 1992, Sutton and Barto, 2018] and policy gradient theorem [Sutton et al., 1999] we can define an estimate of the gradient $\nabla_\theta L(\theta)$ of the non-differentiable return $r$ as:

$$\nabla_\theta \mathcal{L}_{rl}(\theta) \approx -\mathbb{E}_{W^s \sim p_\theta}[r(W^s)\nabla_\theta \log p_\theta(W^s|D)]$$

$$\approx -\mathbb{E}_{W^s \sim p_\theta}[r(W^s)\sum_t \nabla_\theta \log p_\theta(w_t^s|w_{1:t-1}^s, D)] \tag{3}$$

**Defining Return by Unit Test Signals.** For each sample sequence $W^s$, the return $r$ can be defined heuristically by checking its functional correctness. We pass generated programs together with the corresponding unit tests to a compiler. From the outputs of the tests, we can determine the return $r$:

$$r(W^s) = \begin{cases} -1.0 & \text{, if } W^s \text{ cannot be compiled (i.e. compile error)} & (4) \\ -0.6 & \text{, if } W^s \text{ cannot be executed with unit tests (i.e. runtime error)} & (5) \\ -0.3 & \text{, if } W^s \text{ failed any unit test} & (6) \\ +1.0 & \text{, if } W^s \text{ passed all unit tests} & (7) \end{cases}$$

However, in related domains such as text-to-SQL research [Zhong et al., 2018, Xu et al., 2018], we note that this approach to estimate returns can lead to an unstable training process with high variance of the gradient estimate following Eq. (3) with mini-batches in training.

**Return with a Baseline.** In order to alleviate this variance, we adopt a "baseline" [Sutton and Barto, 2018]. Specifically, we use a greedy decoding strategy as a baseline and any generated samples that outperform this baseline are given positive return estimation, and negative return estimation otherwise. This relative normalization technique allows models to explore imperfect programs, as long as their returns are better than the baseline's. Given a training sample, we denote the return of the baseline $r(W^b)$ and the expected gradient is computed as:

$$\nabla_\theta \mathcal{L}_{rl}(\theta) \approx -\mathbb{E}_{W^s \sim p_\theta}[(r(W^s) - r(W^b))\sum_t \nabla_\theta \log p_\theta(w_t^s|w_{1:t-1}^s, D)] \tag{8}$$

Note that at each decoding step $t$, our greedy decoding baseline is independent from the action $w_t^s$ and hence, the expected gradient term $\nabla_\theta \mathcal{L}_{rl}(\theta)$ from Eq. (3) remains the same in Eq. (8).

**Intermediate Return by Critic as Error Predictor.** We observe that the above gradient estimate is only based on a final return at the end of the decoding process. However, programs often follow fixed syntactical rules in which a single token such as an additional white-space character can render a program erroneous. Therefore, Eq. (8) becomes too restrictive. A straightforward solution

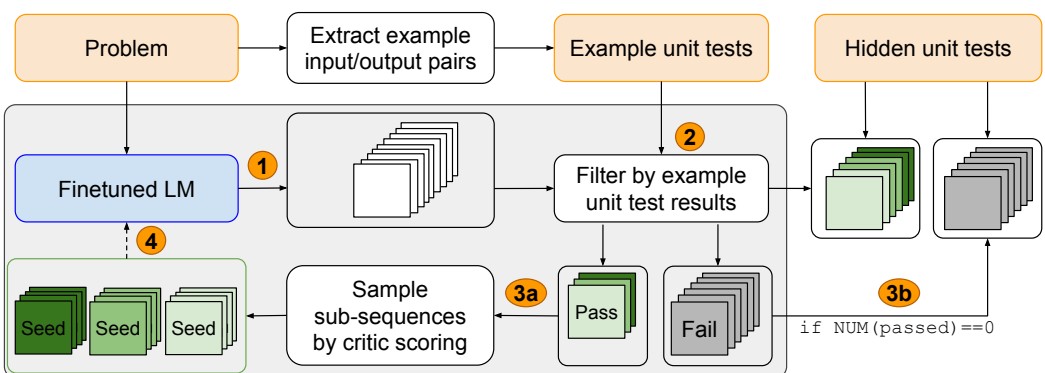

Figure 3: **Overview of our Critic Sampling (CS) approach:** (1) For each test problem, we first use finetuned LM to generate a set of solution programs. (2) From the problem description, we extract example unit tests and test against generated solutions. (3) If there are any passed solutions, we pass them to the critic model to sample sub-sequences. (4) The sub-sequences from (3) are used as seed sequences to condition the LM to regenerate solution programs, repeating the steps from (1) onward.

is to use token-based similarity scores [Papineni et al., 2002, Ren et al., 2020]) between each subsequence $W_{1:t}^s$ and the ground truth. However, code matching is not an ideal return measure due to its poor correlation with program correctness [Hendrycks et al., 2021, Chen et al., 2021a, Austin et al., 2021] which can only be measured against fully complete programs. Alternatively, we introduce a *critic* model (see Appendix **??** for an overview). The critic model is parameterized as a neural network with parameters $\phi$ that receives inputs as the problem description $D$ and a sampled program $W^s = \{w_1^s, \ldots, w_T^s\}$. The critic is trained to infer the unit test outcome; one of $\{\text{CompileError}, \text{RuntimeError}, \text{FailedTest}, \text{PassedTest}\}$ as described in the return definitions in Eq. (4) to (7). The training objective of the critic $\phi$ can be expressed as:

$$\mathcal{L}_{critic}(\phi) = -\log p_\phi(u|W^s, D) \tag{9}$$

where $u$ denotes the ground-truth unit test outcome given by the compiler. We use Transformer models of smaller sizes than the actor model as the base architecture for the critic model. The contextual hidden states of the program tokens $(h_1, \ldots, h_T)$ obtained from the critic model decoder are max-pooled along the sequence length dimension $h^{\text{pool}} = \text{Pooling}(h_1, \ldots, h_T)$. The critic's prediction on the unit test outcome is computed as $\hat{u} = \text{softmax}(\text{Linear}(h^{\text{pool}}))$.

Given a learned critic, we use the probability distribution $\hat{v}_t = \text{softmax}(\text{Linear}(h_t))$ to estimate the token-level value $\hat{q}$ of $w_t^s$ *in relation to* the ground-truth unit test output (note that we use the token level contextual representation $h_t$ here, before the pooling operation). Specifically, $\hat{q}_\phi(w_t^s) = \hat{v}_t[u]$ where $\hat{v}[.]$ denotes the probability of a specific unit test outcome from the four possible ones. We use this estimate to train the actor LM model with intermediate returns:

$$\nabla_\theta \mathcal{L}_{rl}(\theta) \approx -\mathbb{E}_{W^s \sim p_\theta}[(r(W^s) - r(W^b)) \sum_t \hat{q}_\phi(w_t^s) \nabla_\theta \log p_\theta(w_t^s|w_{1:t-1}^s, D)] \tag{10}$$

**Generating Programs with Example Unit Tests and Critic.** We leverage the unit tests provided in the input problem description to improve the generation procedure during inference too (see Figure 3 for an overview and Appendix **??** for step-by-step explanation). For each problem, we generate $N$ programs, out of which we only select programs that pass example tests (leading to a set $\mathcal{F}$) and filter out the rest. To improve sample quality, we perform another round of generation where we use sub-sequences from these filtered samples as prompts (or "seed" sequences) to the actor LM. We employ a separate critic model ($\phi_{\text{test}}$) to guide our choice of sub-sequences from these filtered samples. This critic model is trained with a similar objective as Eq. (9), but in a binary classification setup with $\{\text{FailedTest}, \text{PassedTest}\}$ labels.

Let $W^{\text{filter}} = \{w_1, \ldots, w_T\}$ denote a generated sample that passes the example unit tests. We use the critic model to assign a value to each token $\hat{q}_{\phi_{\text{test}}}(w_t) = p_{\phi_{\text{test}}}(\hat{u} = \text{PassedTest}|w_{1:t}, D)$ corresponding to the critic's predicted probability of the sub-sequence till $t$ passing the unit tests. We split the sequence at position $t_{\text{max}}$ corresponding to the highest critic assigned value and use

the left split as the seed for the next stage. If this seed sequence till $t_{\max}$ contains a token with $p_{\phi_{\text{test}}}(\text{FailedTest}) > p_{\phi_{\text{test}}}(\text{PassedTest})$, we further chop it at this token by removing tokens on the right. This is done to pick prompts that are likely to generate successful programs in the next round.

We use these seeds to initialize and condition the (actor) LM to resample new tokens till we encounter the *<endoftext>* token. In this round, each seed sequence can be stacked $N/|\mathcal{F}|$ times for upsampling. This results in the same number of output programs $N$. We call this generation procedure "Critic Sampling" (CS). We use mini-batch generating to improve efficiency during inference and employ nucleus sampling with a batch size of $N = 200$. While we do incur additional costs to re-sample using the seed sequences, we are only required to generate partial programs in the re-generation stage, making this stage less expensive than conventional generating procedures.

# 4 Experiments

## 4.1 Experimental Setups and Datasets

**Pretraining Setup.** We pretrain a CodeT5-large model (770M) from scratch following T5-large's architecture [Raffel et al., 2020]. We follow the pretraining setups in CodeT5 [Wang et al., 2021] with the modifications as proposed in §3.2. We evaluate this new pretrained CodeT5 model on CodeXGLUE [Lu et al., 2021] and achieve new SOTA results (see Appendix **??**).

**APPS Benchmark.** We choose the challenging APPS program synthesis benchmark [Hendrycks et al., 2021], as it has large coding problems of varying difficulties collected from multiple coding websites. It includes training and test splits, each of which has 5000 samples of programming tasks with diverse levels of difficulty, including "Introductory", "Interview", and "Competition" levels. Each sample includes 20 unit tests on average to validate the functional correctness of programs.

**Finetuning Setup.** Due to the potential large number of trajectories (i.e. $\mathcal{V}^T$) to generate a sequence and the unstable feedback loop between actor and critic [Lillicrap et al., 2015, Wang et al., 2018], we applied imitation learning to first warm-start a pretrained LM model with $\mathcal{L}_{ce}$ only for up to 10 epochs. We then sampled sequences of programs from this actor network to train the critic while keeping the parameters of the actor network frozen. For experiments with CodeT5 actor models, we use the CodeT5-small architecture for the critic model, and GPT2-small critic architecture when the actor models are GPT variants. After training the critic, we then apply both $\mathcal{L}_{ce}$ and $\mathcal{L}_{rl}$ with equal weights to finetune the actor network.

**Evaluation.** We follow [Hendrycks et al., 2021, Chen et al., 2021a] and evaluate the models using the *pass@k* metric, which is the percentage of problems solved by using $k$ generated programs per problem. We also follow Li et al. [2022] and use *n@k* metric which only considers a subset of $n$ candidates from $k$ generated programs per problem. The subset of $n$ candidates are typically selected by a filtering method by passing generated programs through *example tests* given as part of the problem description [Chen et al., 2021a, Li et al., 2022].

For more details of the experimental setup, please refer to Appendix **??**.

## 4.2 Experimental Results on APPS

**Baselines.** As reported by Hendrycks et al. [2021], we compared our models with several baselines, including GPT2 [Radford et al., 2019], GPT-Neo [Black et al.], and GPT3 [Brown et al., 2020]. We also compare the results with Codex [Chen et al., 2021a] and AlphaCode [Li et al., 2022]. Note that by default, results of pretrained LMs (except for Codex and GPT3) are from models finetuned on APPS using the standard loss $\mathcal{L}_{ce}$ only. In our ablations, since CodeRL is model-agnostic, we can also integrate it with GPT variants such as GPT-J [Wang and Komatsuzaki, 2021] and GPT-Neo.

**Overall Results.** Firstly, Table 1a shows that the CodeRL with the CodeT5 model can achieve significant performance gains, outperforming many pretrained LMs of much larger sizes. Specifically, our approach achieved new SOTA results of 2.57% *pass@1*, 6.21% *pass@5*, and 19.36% *pass@1000*. Table 1b shows that when evaluating on a subset of filtered code samples, our CodeRL+CodeT5 can achieve SOTA results of 7.83% *1@k* and 11.61% *5@k*. Note that while CodeRL incurs additional computation cost during inference with CS, our approach only requires much lower $k$ to achieve comparable performance with other models. Specifically, with $k = 1000$ only, our model performance is as good as AlphaCode with a much larger generation budget of $k = 50000$.

Table 1: **Results on APPS:** Overall, CodeRL can bring the performance gains on CodeT5 models and achieves new SOTA on both *pass@k* and *n@k* metrics. "Intro": introductory, "Inter": interview, "Comp": competition-level tasks.

(a) Performance by *pass@k* with $k = \{1, 5, 1000\}$

| Model | Size | pass@1 | | | | pass@5 | | | | pass@1000 | | | |
|---|---|---|---|---|---|---|---|---|---|---|---|---|---|
| | | Intro | Inter | Comp | All | Intro | Inter | Comp | All | Intro | Inter | Comp | All |
| Codex | 12B | 4.14 | 0.14 | 0.02 | 0.92 | 9.65 | 0.51 | 0.09 | 2.25 | 25.02 | 3.70 | 3.23 | 7.87 |
| AlphaCode | 1B | - | - | - | - | - | - | - | - | 17.67 | 5.24 | 7.06 | 8.09 |
| GPT3 | 175B | 0.20 | 0.03 | 0.00 | 0.06 | - | - | - | - | - | - | - | - |
| GPT2 | 0.1B | 1.00 | 0.33 | 0.00 | 0.40 | 2.70 | 0.73 | 0.00 | 1.02 | - | - | - | - |
| GPT2 | 1.5B | 1.30 | 0.70 | 0.00 | 0.68 | 3.60 | 1.03 | 0.00 | 1.34 | 25.00 | 9.27 | 8.80 | 12.32 |
| GPT-Neo | 2.7B | 3.90 | 0.57 | 0.00 | 1.12 | 5.50 | 0.80 | 0.00 | 1.58 | 27.90 | 9.83 | 11.40 | 13.76 |
| GPT-J | 6B | 5.60 | 1.00 | 0.50 | 1.82 | 9.20 | 1.73 | 1.00 | 3.08 | 35.20 | 13.15 | 13.51 | 17.63 |
| CodeRL+CodeT5 | 770M | **6.77** | **1.80** | **0.69** | **2.57** | **15.27** | **4.48** | **2.36** | **6.21** | **38.10** | **14.33** | **15.70** | **19.36** |

(b) Performance by *n@k* with $k$ up to 50000 and $n = \{1, 5\}$

| Model | Size | $k$ | 1@k | | | | 5@k | | | |
|---|---|---|---|---|---|---|---|---|---|---|
| | | | Intro | Inter | Comp | All | Intro | Inter | Comp | All |
| Codex | 12B | 1000 | **22.78** | 2.64 | 3.04 | 6.75 | **24.52** | 3.23 | 3.08 | 7.46 |
| AlphaCode | 1B | 1000 | - | - | - | - | 14.36 | 5.63 | 4.58 | 7.17 |
| AlphaCode | 1B | 10000 | - | - | - | - | 18.18 | 8.21 | 6.65 | 9.89 |
| AlphaCode | 1B | 50000 | - | - | - | - | 20.36 | **9.66** | 7.75 | 11.42 |
| CodeRL+CodeT5 | 770M | 1000 | 16.52 | **6.16** | **4.15** | **7.83** | 24.49 | 8.58 | **7.82** | **11.61** |

Table 2: **Ablation results with variants of return estimates:** CodeT5 model that is trained with return estimates using a baseline ($W_b$) and a critic-based return estimates ($\hat{q}_\theta$) can achieve the best performance. "dist." indicates a rule-based approach that estimates returns following a linear decay by token positions from $t = 1$ to $t = T$.

| # | $W^b$ | $\hat{q}_\phi$ | pass@1 | | | | pass@5 | | | |
|---|---|---|---|---|---|---|---|---|---|---|
| | | | Intro | Inter | Comp | All | Intro | Inter | Comp | All |
| A | ✓ | - | 4.60 | 1.10 | 0.20 | 1.62 | 7.10 | 1.57 | 0.40 | 2.44 |
| B | - | ✓ | 4.00 | 0.87 | 0.20 | 1.36 | 5.60 | 1.30 | 0.20 | 1.94 |
| C | ✓ | dist. | 4.90 | 1.03 | 0.20 | 1.64 | 7.80 | 1.60 | 0.30 | 2.58 |
| D | ✓ | ✓ | **6.20** | **1.50** | **0.30** | **2.20** | **9.39** | **1.90** | **0.42** | **3.10** |

## 4.3 Ablation Studies

In this section, for a fair comparison between variants of return estimates and learning objectives, we report the results of *pass@k* where $k = \{1, 5\}$ with beam search decoding. For larger $k$, we report the results with and without the CS procedure.

**Impacts of Return Estimates.** Table 2 show the results of CodeT5-770M trained by different approaches to estimate returns of code samples. Overall, we report that the CodeRL objective with relative token-level return estimates by our critic model (Model D) can achieve the best performance on *pass@1* and *pass@5*. Secondly, we note that using absolute returns without a baseline (Model B) could lead to the most performance drop, as this approach heavily penalizes all incorrect samples (even though they might still be better than a naive baseline). Thirdly, without a critic model, simply assigning identical rewards to all tokens in a code sample (Model A) is disadvantageous as these return estimates are too restrictive to be used as feedback signals for RL training. Finally, we experimented with a distance-based critic which assumes that token values decay linearly from $t = 1$ to $t = T$ (Model C). The lower performance suggests the benefit of training a critic network to compute the returns rather than relying on rule-based approaches.

**Impacts of Learning Objectives.** Table 3 shows the results with different combinations of $\mathcal{L}_{ce}$ and $\mathcal{L}_{rl}$. We experiment with using only $\mathcal{L}_{rl}$ and note the problem of vanishing gradients during finetuning [Ranzato et al., 2016, Bahdanau et al., 2016]. Secondly, we note that by using only $\mathcal{L}_{ce}$ for further finetuning, despite improvement in losses during training time, the model performance indeed degrades during test time. We expect these models are overfitting to the training data. Interestingly, a naive approach of $\mathcal{L}_{ce}$ with synthetic samples $W^s$, all of which are treated as correct codes with $r(W^s) = 1$, still leads to some performance improvement with GPT-Neo on *pass@5* (but not in

Table 3: **Ablation results with different learning objectives:** We experiment with both CodeT5 and GPT-Neo with different combinations of cross-entropy loss $\mathcal{L}_{ce}$ and reinforcement learning loss $\mathcal{L}_{rl}$. Note that these losses are applied on models that are already warm-started with conventional cross-entropy losses for up to 10 epochs.

| $\mathcal{L}_{ce}$ | $\mathcal{L}_{rl}$ | pass@1 | | | | pass@5 | | | |
|---|---|---|---|---|---|---|---|---|---|
| | | Intro | Inter | Comp | All | Intro | Inter | Comp | All |
| | | | | GPT-Neo | | | | | |
| - | - | 3.90 | 0.57 | 0.00 | 1.12 | 5.50 | 0.80 | 0.00 | 1.58 |
| ✓ | - | 2.70 | **0.90** | 0.10 | 1.10 | 5.00 | 1.43 | 0.30 | 1.92 |
| ✓(+$W^s$) | - | 2.90 | 0.80 | **0.30** | 1.12 | 5.20 | **1.57** | **0.40** | 2.06 |
| - | ✓ | 3.30 | 0.80 | 0.20 | 1.18 | 5.30 | **1.57** | 0.20 | 2.04 |
| ✓ | ✓ | **4.70** | 0.73 | **0.30** | **1.44** | **6.58** | 1.54 | 0.18 | **2.28** |
| | | | | CodeT5-770M | | | | | |
| - | - | **6.60** | 1.03 | 0.30 | 2.00 | 8.80 | 1.67 | **0.70** | 2.90 |
| ✓ | - | 4.60 | 0.93 | 0.10 | 1.50 | 7.00 | 1.37 | 0.20 | 2.26 |
| ✓(+$W^s$) | - | 5.10 | 1.10 | 0.40 | 1.76 | 8.30 | 1.43 | **0.70** | 2.66 |
| - | ✓ | 5.00 | 0.90 | **0.50** | 1.64 | 7.60 | 1.53 | 0.60 | 2.56 |
| ✓ | ✓ | 6.20 | **1.50** | 0.30 | **2.20** | **9.39** | **1.90** | 0.42 | **3.10** |

Table 4: **Ablation results of Critic Sampling (CS):** Overall, using CS can lead to higher passing rates as program generations are conditioned on seed sequences filtered by their test results.

| Metric | Approach | Intro | Inter | Comp | All |
|---|---|---|---|---|---|
| pass@200 | without CS | 26.79 | 8.73 | 7.60 | 12.12 |
| | with CS | 29.10 | 9.67 | 9.50 | 13.52 |
| pass@1000 | without CS | 35.30 | 13.33 | 13.60 | 17.78 |
| | with CS | 38.10 | 14.33 | 15.70 | 19.36 |
| 1@1000 | without CS | 16.27 | 6.00 | 4.27 | 7.71 |
| | with CS | 16.52 | 6.16 | 4.15 | 7.83 |

other cases). Finally, we found that using both $\mathcal{L}_{ce}$ and $\mathcal{L}_{rl}$ results in a more consistent performance improvement overall on *pass@1* and *pass@5* for the GPT-Neo and CodeT5 models.

**Impact of Critic Sampling** . Table 4 shows the ablation results of critical sampling (CS) during inference. Overall, we found positive impact of CS for improving *pass@200* and *pass@1000* metrics. In addition, the positive impacts of critic sampling on *pass@1* and *pass@5* are indicated by comparing the results of Table 1a and Table 2 (row D, in which we only used conventional beam search decoding without critic sampling). We can observe that, using critic sampling, model performance increases from 2.2% *pass@1* (3.1% *pass@5*) to 2.57% *pass@1* (6.21% *pass@5*). Interestingly, from Table 4, we observe that CS does not provide a significant gain on the $n@k$ metric. Note that $n@k$ measures the solving rate among the subset $\mathcal{F}$ filtered from $k$ samples. As CS will technically increase the size of this subset, the $n@k$ metric will consider an exponentially larger number of options of $n$ samples than before. This will normalize $n@k$ by a larger pool of $n$ candidate set, resulting in less impact of CodeRL on the results. We recommend additional post-processing steps such as candidate ranking [Cobbe et al., 2021] to improve the $n@k$ performance.

**Impacts of Pretraining Approaches for CodeT5.** Table 5 reports the results of CodeT5 with different configurations of model sizes, pretraining data, and pretraining objectives. For a fair comparison, all models are only finetuned with $\mathcal{L}_{ce}$ on APPS. As observed in prior work [Chen et al., 2021a, Austin et al., 2021], scaling up the number of model parameters or the size of the pretraining data can significantly improve the model performance of downstream synthesis tasks. We also find that enhancing the pretraining objectives with next token prediction (NTP) is vital for generation tasks, surpassing just masked span prediction (MSP) from the original CodeT5.

## 4.4 Zero-shot Evaluation on MBPP Benchmark

Finally, we test the zero-shot transfer ability of CodeRL on another smaller and simpler program synthesis benchmark MBPP [Austin et al., 2021].

Table 5: **Ablation results of CodeT5 pretrained model variants:** We report the results of models pretrained on different configurations by model size, pretraining data, and pretraining task. CSN: CodeSearchNet, GCPY: Github Code Python, MSP: Masked Span Predition, NTP: Next Token Prediction. For a fair comparison, all models are finetuned only with $\mathcal{L}_{ce}$ on APPS.

| Size | Data | Task | pass@1 | | | | pass@5 | | | |
|------|------|------|-------|-------|------|------|-------|-------|------|------|
| | | | Intro | Inter | Comp | All | Intro | Inter | Comp | All |
| 60M | CSN | MSP | 1.40 | 0.67 | 0.00 | 0.68 | 2.60 | 0.87 | 0.10 | 1.06 |
| 220M | CSN | MSP | 2.50 | 0.73 | 0.00 | 0.94 | 3.30 | 1.10 | 0.10 | 1.34 |
| 770M | CSN | MSP | 3.60 | 0.90 | 0.20 | 1.30 | 4.30 | 1.37 | 0.20 | 1.72 |
| 770M | +GCPY | MSP | 4.30 | **1.10** | 0.20 | 1.56 | 5.60 | 1.47 | 0.30 | 2.06 |
| 770M | +GCPY | +NTP | **6.60** | 1.03 | **0.30** | **2.00** | **8.80** | **1.67** | **0.70** | **2.90** |

Table 6 reports the results of our CodeRL+CodeT5 on MBPP benchmark compared with finetuned GPT models of up to 137B size. Our CodeRL+CodeT5 (ZS) was trained on APPS and then evaluated on MBPP in a zero-shot setting. We observe that CodeRL with CodeT5 of a much smaller model size yields surprisingly good zero-shot performance, setting a new SOTA result of 63.0% *pass@80* over GPT-137B's 61.4% *pass@80*. This validates the strong zero-shot transfer ability of CodeRL for unseen tasks.

For additional experiments, analysis, and qualitative results, please refer to Appendix **??**.

Table 6: **Results on MBPP:** we test the zero-shot transfer ability of CodeRL. CodeRL+CodeT5 (ZS) which was trained on APPS with $\mathcal{L}_{rl}$ and evaluated on MBPP [Austin et al., 2021] in a zero-shot setting, achieves new SOTA.

| Model | Size | pass@80 |
|-------|------|---------|
| GPT | 224M | 7.2 |
| GPT | 422M | 12.6 |
| GPT | 1B | 22.4 |
| GPT | 4B | 33.0 |
| GPT | 8B | 40.6 |
| GPT | 68B | 53.6 |
| GPT | 137B | 61.4 |
| CodeRL+CodeT5 (ZS) | 770M | **63.0** |

## 5   Limitations and Broader Impacts

One limitation of our approach is the computation cost of training critic model to estimate returns in addition to the original LM (actor network). However, in practice, we found that training a good critic model does not require large-scale models to attain a decent performance. For instance, a finetuned critic model initialized from a pretrained GPT-2 (small) can achieve over 75% error prediction accuracy on synthetic samples.

Program synthesis can lead to substantial positive social impacts, e.g., transforming future software developing tools, increasing the productivity of developers, and improving the accessibility and quality of programming courses. Yet, some risks and bias issues are still worth considering before deploying such models at scale. For example, training data from public GitHub code repos may contain vulnerabilities and the resulting synthesis models may generate programs with weak security measures [Hammond Pearce et al., 2021].

## 6   Conclusion

We present CodeRL, a novel framework for program synthesis, using deep reinforcement learning to improve pretrained LMs, by exploiting unit test signals in both training and inference stages. Specifically, we introduce an actor-critic training approach to optimize pretrained LMs with dense feedback signals on synthetic code samples. During inference, we propose a new generation procedure with critical sampling, which enables the model to automatically regenerate programs based on feedback from unit tests and critic scores. We integrate CodeRL with the improved CodeT5-large model (770M) and achieve new SOTA results on both the APPS and MBPP benchmarks, surpassing the prior SOTA by massive pretrained LMs of much larger model sizes. Our comprehensive analysis shows that CodeRL achieved consistent improvement upon the conventional pretrained LMs for code generation tasks. CodeRL is a general framework that integrates pretrained LMs and RL holistically for program synthesis, and can be extended and improved in various ways. For example, it can be easily integrated with other better pretrained LMs and can be improved with more fine-grained feedback from the environment, such as feedback received from a static code analyzer.

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
