# CodeRL: Mastering Code Generation through Pretrained Models and Deep Reinforcement Learning Supplementary

## A Overview of Critic Model

Figure 1 shows an overview of our critic model. In our CodeRL framework, besides the actor LM network $\theta$, we introduce a critic model that is trained as an error predictor and parameterized as a neural network with parameters $\phi$. The critic receives as inputs a problem description $D$ and a corresponding synthetic program $W^S$ sampled from the actor. The critic is required to output a prediction of the unit test outcome of the input program. We define 4 possible outcomes $u$: *CompileError*, *RuntimeError*, *FailedTest*, and *PassedTest*. The critic model is trained by minimizing the following loss:

$$\mathcal{L}_{critic}(\phi) = -\log p_\phi(u|W^s, D) \tag{1}$$

The ground-truth outcome of a synthetic sample is obtained by passing it to the unit tests corresponding to the problem. Note that since our critic model is applied in a supervised learning environment with available ground truth, we also use the training samples from the original dataset with ground truth output $u = $ *PassedTest* to train the critic.

The learned hidden state representations of program tokens when passed through the critic are then used to measure their return estimates for our RL optimization objective. The return estimates are incorporated as intermediate returns at decoding steps to compute the expected gradient of the actor network $\nabla_\theta \mathcal{L}_{rl}(\theta)$.

## B Critic Sampling Procedure

Refer to Algorithm 1 for a step-by-step explanation of our critic sampling procedure.

## C Additional Experimental Setup Details

**Pretraining Setup.** For CodeT5, we adopt the code-specific tokenizer as described by Wang et al. [2021]. Note that we employ 6 programming languages (PLs) in CodeSearchNet [Husain et al., 2019] (CSN) instead of 8 PLs in CodeT5 as C/C# datasets are not publicly available. We employ only the pretraining task of masked span prediction (MSP) in CodeT5 and hence, we do not have to parse programs into abstract syntax trees (ASTs) to obtain the identifier information. This preprocessing step was required in other original pretraining tasks like masked identifier prediction [Wang et al., 2021]. To further speed up training, we concatenate data samples to batch size 512 for pretraining with MSP and the resulting number of tokens is 1.1B.

Submitted to 36th Conference on Neural Information Processing Systems (NeurIPS 2022). Do not distribute.

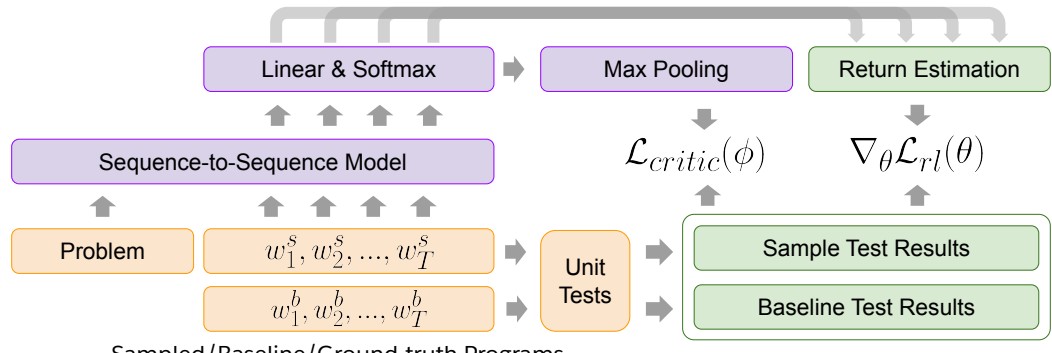

Figure 1: **Overview of our critic model:** The critic receives problem specifications and programs as inputs. For each program, the critic predicts one of four possible test outcomes: $\{CompileError, RuntimeError, FailedTest, PassedTest\}$. The learned hidden state representations are then used to estimate the returns of synthetic samples to finetune the actor network. To improve and stabilize the training process, baseline programs are used to compute relative return estimates.

**APPS Benchmark.** We follow the same preprocessing step as in [Hendrycks et al., 2021] to formulate the input sequences from problem descriptions. APPS consists of 10,000 coding problems with a 50-50 train-test split. Each problem is accompanied by 23.2 correct Python programs and 21.2 unit tests on average. The average length per problem is 293.2 words and the average length per program is 18.0 lines. The dataset is categorized into three levels of difficulty: Introductory (3639, train/test=2639/100), Interview (5000, train/test=2000/3000), and Competition (1361, train/test=361/1000). Similarly to [Hendrycks et al., 2021], we employ the strict accuracy metric to evaluate the functional correctness of a program, where it is counted as correct if it can pass all the unit tests corresponding to the problem.

**MBPP Benchmark.** We additionally include another smaller and simpler Python program synthesis dataset called MBPP [Austin et al., 2021] (Mostly Basic Programming Problems) for evaluation. The dataset contains 974 instances with 374/90/500 instances for training/validation/testing respectively and 10 reserved for few-shot learning. The problems are typically short, usually one sentence of natural language descriptions each. Each problem is accompanied by 1 correct solution (6.8 lines of code on average) and 3 unit tests in the form of `assert` statements for validating the functional correctness. Unlike APPS, unit tests in MBPP are not hidden and are explicitly incorporated into the source sequences for program synthesis models. This might encourage models to be overfitting to these `assert` statements via hard-coding an if-expression very occasionally. However, for a fair comparison with the baselines, we construct the source sequences in the same way as in prior work. Specifically, we adopt the same prompt format as in [Austin et al., 2021] to prepare the input sequence as: problem descriptions + "Your code should satisfy these tests:" + 3 assert statements.

**Finetuning Setup.** Following [Bahdanau et al., 2016], since our RL method is applied in a supervised learning task, in addition to synthetic programs, we also use the ground-truth programs of training samples to train the critic. These samples are considered perfect programs and always have a label of *PassedTest*. To optimize the LM actor network, in practice, following previous work [Bahdanau et al., 2016, Rennie et al., 2017, Wang et al., 2018], in each training optimization step, we can simply approximate the expected gradient with a single sample $W_s \sim p_\theta$:

$$\nabla_\theta \mathcal{L}_{rl}(\theta) \approx -(r(W^s) - r(W^b)) \sum_t \hat{q}_\phi(w_t^s) \nabla_\theta \log p_\theta(w_t^s | w_{1:t-1}^s, D) \tag{2}$$

**Configurations.** For pretraining, we perform our experiments on a Kubernetes with 16 A100-40G GPUs on Google Cloud Platform and the total pretraining duration is around 21 days. In the first pretraining stage with MSP, we employ a corruption rate of 15%, a peak learning rate (LR) of 2e-4,

**Algorithm 1:** Critic sampling procedure to generate programs

**Input:** Problem $D$, Language model $\theta$, Critic model $\phi$
**Output:** A set of $N$ generated solution programs

1 **begin**
    /* step 1:  generate program solutions by LM */
2     **a set of output solutions** $\mathcal{S} \longleftarrow \emptyset$
3     **for** $N$ **times do**
        /* use sampling decoding to generate programs */
4         **Program** $\hat{W}_i \longleftarrow$ Sampling $p_\theta(W_i|D) \longleftarrow$ Sampling $p_\theta(w_t^i|w_{1:t-1}^i, D)$
5         $\mathcal{S} \longleftarrow \mathcal{S} \cup \{\hat{W}_i\}$
    /* step 2:  extract example unit tests from problem */
6     **example unit tests** $\mathcal{I} \longleftarrow$ ExtractExampleInputOutput$(D)$
    /* step 3:  filter for passed programs and pass to the critic model to select sub-sequences */
7     **a set of filtered programs** $\mathcal{F} \leftarrow \emptyset$
8     **for each generated program** $\hat{W}_i \in \mathcal{S}$ **do**
9         **Testoutcomes** $u_i \longleftarrow$ RunTests$(\hat{W}_i, \mathcal{I})$
10         **if** $u_i =$ "PassedTest" **then**
11             $\mathcal{F} \longleftarrow \mathcal{F} \cup \{\hat{W}_i\}$

    /* If no passed samples, return all current programs for evaluation */
12     **if** $|\mathcal{F}| = 0$ **then return** $\mathcal{S}$
13     **upsampling param** $N^{'} = N/|\mathcal{F}|$
14     **a set of output solutions** $\mathcal{S}^{'} \longleftarrow \emptyset$
15     **for each filtered program** $\hat{W}_i \in \mathcal{F}$ **do**
16         **subsequence** $W_i^{sub} \longleftarrow$ Sampling $p_\phi(u_i = \text{PassedTest}|\hat{W}_i, D)$
17         $\longleftarrow$ Sampling $p_\phi(u_i = \text{PassedTest}|\hat{w}_{1:t-1}, D)$
        /* step 4:  subsequence as seeds to regenerate programs */
18         **length of subsequence** $m \longleftarrow |W_i^{sub}|$
19         **for** $N^{'}$ **times do**
20             $\hat{W}_j \longleftarrow$ Sampling $p_\theta(W_j|W_i^{sub}, D) \longleftarrow$ Sampling $p_\theta(w_t^j|w_{m:t-1}^j, W_i^{sub}, D)$
21             $\mathcal{S}^{'} \longleftarrow \mathcal{S}^{'} \cup \{\hat{W}_j\}$

    /* return the regenerated programs for evaluation */
22     **return** $\mathcal{S}^{'}$

---

and a batch size of 2048. We pretrain on CSN for 150 epochs (10 days) and then on GCPY for 10 epochs (5 days). For the second stage pretraining with NTP, we adopt a peak LR of 1e-4 and a batch size of 256, and pretrain for 10 epochs (6 days). We set the maximum length to 768 and 600 for source and target sequences respectively for this objective. For all experiments, we employ an AdamW optimizer [Loshchilov and Hutter, 2019] with a 0.05 weight decay and a linear decay LR scheduler with a warmup step of 1000.

For finetuning on APPS, we adopt a batch size of 64 and warmup LR from 0 to 2e-5 for the first 500 steps and polynomially (power=0.5) decay to 1e-5 until the end of 10 epochs, which takes around 30 hours on one A100 GPU. We set the maximum source and target sequence length to 600 and 512 respectively.

To train the critic model (either a GPT2-small or a CodeT5-base), we train the models with synthetic sampled programs for up to 10 epochs. We found that in this phase, the critic model usually converges quite earlier than the 10-epoch training limit. Once the critic model is trained, we continue to finetune

Table 1: **CodeT5 results on CodeXGLUE:** Code-to-Text generation results (smoothed BLEU-4) of CodeT5 pretrained with larger data, improved learning objectives, and larger model size

| Model | Ruby | JavaScript | Go | Python | Java | PHP | Overall |
|---|---|---|---|---|---|---|---|
| RoBERTa | 11.17 | 11.90 | 17.72 | 18.14 | 16.47 | 24.02 | 16.57 |
| CodeBERT | 12.16 | 14.90 | 18.07 | 19.06 | 17.65 | 25.16 | 17.83 |
| DOBF | - | - | - | 18.24 | 19.05 | - | - |
| PLBART | 14.11 | 15.56 | 18.91 | 19.30 | 18.45 | 23.58 | 18.32 |
| CoTexT | 14.02 | 14.96 | 18.86 | 19.73 | 19.06 | 24.58 | 18.55 |
| CodeT5-small | 14.87 | 15.32 | 19.25 | 20.04 | 19.92 | 25.46 | 19.14 |
| CodeT5-base | 15.24 | 16.16 | 19.56 | 20.01 | 20.31 | 26.03 | 19.55 |
| CodeT5-large | **15.58** | **16.17** | **19.69** | **20.57** | **20.74** | **26.49** | **19.87** |

Table 2: **CodeT5 results on CodeXGLUE:** Text-to-Code generation results of CodeT5 pretrained with larger data, improved learning objectives, and larger model size

| Model | EM | BLEU-4 | CodeBLEU |
|---|---|---|---|
| GPT-2 | 17.35 | 25.37 | 29.69 |
| CodeGPT-2 | 18.25 | 28.69 | 32.71 |
| CodeGPT-adapted | 20.10 | 32.79 | 35.98 |
| PLBART | 18.75 | 36.69 | 38.52 |
| CoTexT | 20.10 | 37.40 | 40.14 |
| UniXcoder | 22.60 | 38.23 | - |
| CodeT5-small | 21.55 | 38.13 | 41.39 |
| CodeT5-base | 22.30 | 40.73 | 43.20 |
| CodeT5-large | **22.65** | **42.66** | **45.08** |

Table 3: **CodeT5 results on CodeXGLUE:** Code-to-Code generation results of CodeT5 pretrained with larger data, improved learning objectives, and larger model size

| Model | Java to C# | | C# to Java | | Refine Small | | Refine Medium | |
|---|---|---|---|---|---|---|---|---|
| | BLEU-4 | EM | BLEU-4 | EM | BLEU-4 | EM | BLEU-4 | EM |
| Naive copy | 18.54 | 0.00 | 18.69 | 0.00 | 78.06 | 0.00 | 90.91 | 0.00 |
| Roborta (code) | 77.46 | 56.10 | 71.99 | 57.90 | 77.30 | 15.90 | 90.07 | 4.10 |
| CodeBERT | 79.92 | 59.00 | 72.14 | 58.00 | 77.42 | 16.40 | 91.07 | 5.20 |
| GraphCodeBERT | 80.58 | 59.40 | 72.64 | 58.80 | **80.02** | 17.30 | **91.31** | 9.10 |
| PLBART | 83.02 | 64.60 | 78.35 | 65.00 | 77.02 | 19.21 | 88.50 | 8.98 |
| CoTexT | - | - | - | - | 77.79 | 21.03 | 88.40 | 13.11 |
| NSEdit | - | - | - | - | 71.06 | **24.04** | 85.72 | 13.87 |
| CodeT5-small | 82.98 | 64.10 | 79.10 | 65.60 | 76.23 | 19.06 | 89.20 | 10.92 |
| CodeT5-base | **84.03** | 65.90 | **79.87** | 66.90 | 77.43 | 21.61 | 87.64 | 13.96 |
| CodeT5-large | 83.56 | **66.00** | 79.77 | **67.00** | 77.38 | 21.70 | 89.22 | **14.76** |

the actor language model for 10 more epochs using both the conventional NTP loss and RL-based loss.

For MBPP, in finetuning experiments with CodeT5, due to its small training set, we finetune the models for 60 epochs with a constant LR of 2e-5 and a batch size of 32, which takes less than 30 mins on one A100. We set its maximum source and target length to 382 and 306 respectively.

**Critic Sampling.** Note that in CS, many rounds of regeneration can be run to keep improving the output programs. However, in practice, we found that one round of regeneration is sufficient to obtain good programs. In the initial generation, if no samples pass example unit tests, we do not apply critic sampling and simply use all available programs for evaluation.

Table 4: **Ablation results on MBPP:** Ablation results of CodeRL with different CodeT5 model variants with different sizes, pretraining data and objectives on MBPP. CodeT5$^{\dagger}$ is finetuned on APPS and evaluated on MBPP in a zero-shot setting.

| Model | Size | Data | Objective | *pass@80* | *pass@1000* |
|---|---|---|---|---|---|
| GPT finetuned results | | | | | |
| GPT | 224M | Web Doc | LM | 7.2 | - |
| GPT | 422M | Web Doc | LM | 12.6 | - |
| GPT | 1B | Web Doc | LM | 22.4 | - |
| GPT | 4B | Web Doc | LM | 33.0 | - |
| GPT | 8B | Web Doc | LM | 40.6 | - |
| GPT | 68B | Web Doc | LM | 53.6 | - |
| GPT | 137B | Web Doc | LM | 61.4 | - |
| CodeT5 finetuned results | | | | | |
| CodeT5 | 60M | CSN | MSP | 19.2 | 36.2 |
| CodeT5 | 220M | CSN | MSP | 24.0 | 42.8 |
| CodeT5 | 770M | CSN | MSP | 32.4 | 47.8 |
| CodeT5 | 770M | +GCPY | MSP | 34.6 | 51.6 |
| CodeT5 | 770M | +GCPY | +NTP | 46.8 | 66.2 |
| CodeRL zero-shot results | | | | | |
| CodeT5$^{\dagger}$ | 770M | +GCPY | +NTP | 60.2 | 78.4 |
| +CodeRL | 770M | +GCPY | +NTP | 63.0 | 81.8 |

# D  Additional Experimental Results

## D.1  CodeXGLUE Benchmark Results

To validate the effectiveness of our simplified pretraining strategies of CodeT5-large, we extensively evaluate it on a variety of generation tasks in CodeXGLUE [Lu et al., 2021], including code-to-text generation (i.e. summarization, see Table 1), text-to-code generation (see Table 2), and code-to-code generation (i.e., code translation and code refinement, see Table 3). Different from APPS [Hendrycks et al., 2021] and MBPP [Austin et al., 2021], we follow the default similarity-based evaluation metrics in the CodeXGLUE benchmark, including BLEU [Papineni et al., 2002] and CodeBLEU [Ren et al., 2020], and exact match (EM) scores. Table 1, 2, and 3 show that our simplified pretrained CodeT5-large sets new SOTA results on a large majority of the tasks, and hence, can be served as a better foundation model for other code-related generation tasks. Note that in these experiments, we employ the conventional finetuning objective with $\mathcal{L}_{ce}$ and there might be potential to improve the performance further with our CodeRL framework.

## D.2  MBPP Benchmark Results

Following Austin et al. [2021], we adopt temperature sampling to generate multiple candidate solutions. We empirically find that CodeT5 benefits from a higher temperature of $1.2$ (less greedy decoding or more diverse) than their GPT's temperature of $0.5$ on this benchmark.

Table 4 reports the *pass@80* and *pass@1000* results for both finetuned and zero-shot settings. For baselines, we compared with GPT models with sizes ranging from 224M to 137B [Austin et al., 2021], which are pretrained on 2.93B web documents (13.8M containing source code) using standard language modeling objective. Results of GPT models are obtained from the original authors. From the comparison among various CodeT5 variants, we again confirm that larger model sizes and pretraining data, and better pretraining objective of NTP all lead to a performance boost. Particularly, our CodeT5-770M yields a *pass@80* of 46.8%, surpassing GPT-8B's 40.6% with a much smaller model size. In addition, we find CodeT5 models finetuned on APPS can achieve a surprisingly good zero-shot performance on MBPP with a *pass@80* of 60.2% and further improved to 63.0% with the help of CodeRL, which even outperforms the largest GPT-137B's performance of 61.4%. This indicates APPS is a comprehensive program synthesis benchmark and CodeT5+CodeRL models trained on it are able to generalize to other simpler coding tasks. If we further increase the budget of

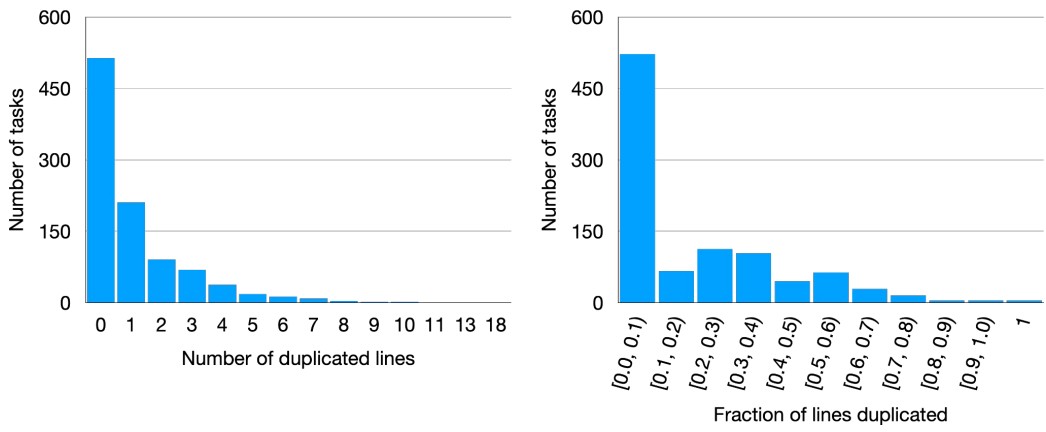

Figure 2: **Analysis of duplicated lines between MBPP and APPS:** The overlap of data between APPS and MBPP appears to be minimal, with only $12.6\%$ MBPP programs with $> 50\%$ lines duplicated in APPS training data.

attempts up to 1000, all models witness a consistent and significant boost in solving rate, especially our CodeT5+CodeRL yielding a new SOTA result of 81.8% *pass@1000*.

A common concern about transfer learning is that the source (APPS) and target (MBPP) tasks might overlap in their training data, which could result in the source model tending to memorize these substantially similar data when applied to the target task. To address this concern, we analyze how many lines of code appear in both the training set of APPS and the programs of MBPP following Austin et al. [2021]. For this analysis, we discard code comments and normalize the whitespaces for each line, and then exclude lines that appear more than twice anywhere in MBPP, as these are likely to be common Python keywords such as `return` and `break`.

Figure 2 illustrates the number of absolute duplicated lines (Left) and the relative fraction of duplicated lines (Right) in the MBPP programs. As can be seen, the overlap between APPS and MBPP seems to be minimal. Only 12.6% MBPP programs have more than half of their lines matched somewhere in the APPS training data. Besides, more than half (514 out of 974) of programs have a zero overlap and 90.9% have only no more than 3 lines overlapping with the APPS training set. If we further require the lines to be consecutive, there are no more than 2 consecutive duplicated lines.

### D.3 APPS Benchmark Results on Competition-level Tasks

Figure 3 shows the results of *pass@k* and *n@k* with $k$ ranging from 1 to 200 and $n = \{1, 5\}$, for CodeRL+CodeT5 and CodeT5 only. We choose to investigate a subset of the APPS test split, which contains the test samples of the highest difficulty level (i.e. competition programming tasks). Since CodeRL is model-agnostic, we also integrate it with GPT-J [Wang and Komatsuzaki, 2021] and report the results. To focus on the impact of our RL optimization, during test time, we compare models using only nucleus sampling and without the CS procedure. Figure 3 shows that the performance gains are quite consistent on both GPT-J and CodeT5. In particular, as $k$ increases, the performance gain of CodeRL is more significant on both GPT-J and CodeT5 models. We attribute these gains to the CodeRL learning objective $\mathcal{L}_{rl}$ that encourages models to explore code solutions drawn from the model's sampling distribution. During test time with an increasing $k$ sampling budget, models are allowed to generate diverse code solutions and the impact of $\mathcal{L}_{rl}$ becomes more significant.

### D.4 CodeT5 Ablation by Training Epochs

Figure 4 shows the performance of CodeT5 model variants by finetuning epochs and by difficulty levels of programming tasks. Note that in these experiments, we only need to compare among CodeT5 model variants by pretraining strategies, and hence, only engage $\mathcal{L}_{ce}$ (imitation learning) in

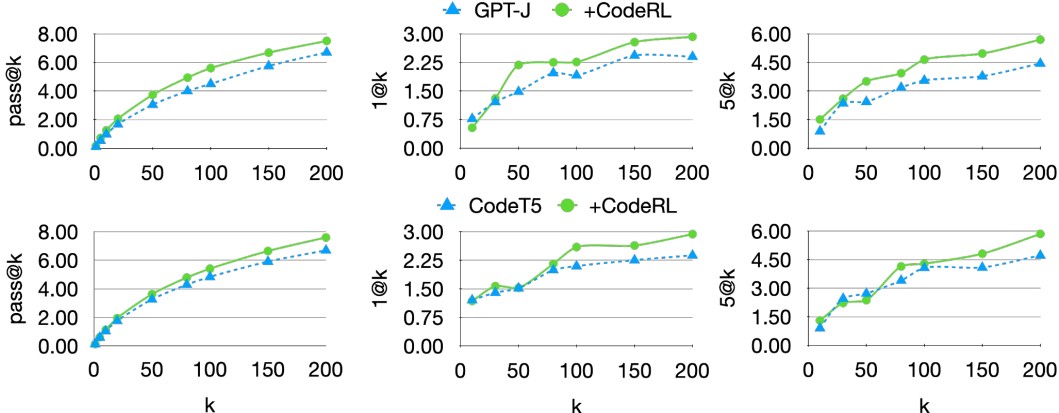

Figure 3: **Results on APPS competition-level test samples:** We investigate the most challenging programming problem tasks, i.e. competition level, in the APPS benchmark. Integrating CodeRL with both CodeT5 and GPT-J, we observe good performance improvement across $pass@k$ and $n@k$ metrics, with increasing performance gains as $k$ increases.

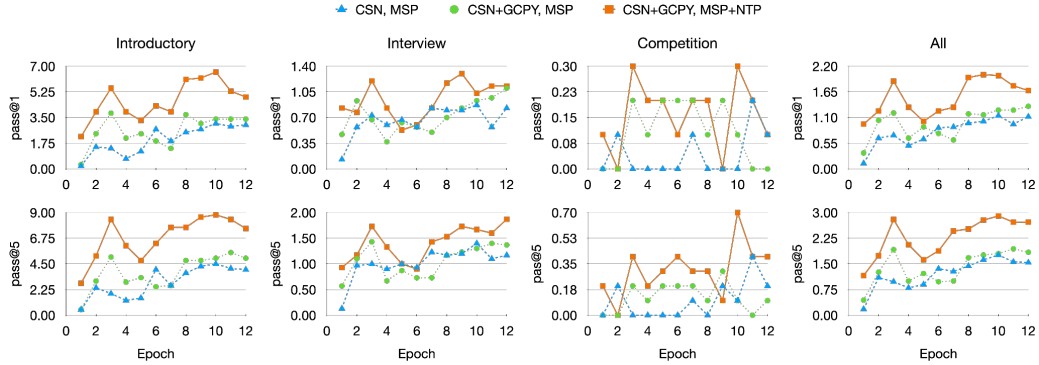

Figure 4: **Ablation results by finetuning epochs**: We report the finetuning progress of CodeT5-770M models that are pretrained on different configurations by pretraining data and pretraining tasks. CSN: CodeSearchNet, GCPY: Github Code Python, MSP: Masked Span Prediction, NTP: Next Token Prediction. All models are finetuned only with $\mathcal{L}_{ce}$ on APPS.

the finetuning stage on APPS. Consistent with our prior analysis (see Section **??** of the main paper), enhancing both pretraining data (with larger data of GCPY) and pretraining objectives (with NTP objective) improves model performance across training epochs in general. Moreover, as noted by our analysis of learning objectives, only using $\mathcal{L}_{ce}$ often leads to overfitting performance, typically after epoch 10 in our case. Hence, to further finetune large-scale LMs, we recommend adopting our RL objective $\mathcal{L}_{rl}$ to utilize synthetic training samples and avoid overfitting models.

## D.5 Impacts of CodeRL on Program Quality by Unit Test Signals

Figure 5 demonstrates the average percentages of generated programs by their test signals. Specifically, we use CodeT5 or CodeRL+CodeT5 to generate programs and randomly select 200 generated programs per test sample in the APPS test split. We pass programs to either example unit tests or hidden unit tests corresponding to the problem and group the output programs by their output signals, including CompileError, RuntimeError, FailedTest, and PassedTest. We observe that integrating CodeRL can increase the likelihood that a program can pass unit tests, and reduces the likelihood that it fails one or more unit tests, or whether it contains compiling or runtime errors. However, we note that there are significant gaps in performance by test signals between example unit tests and hidden unit tests. This observation suggests that example tests are not as comprehensive as hidden

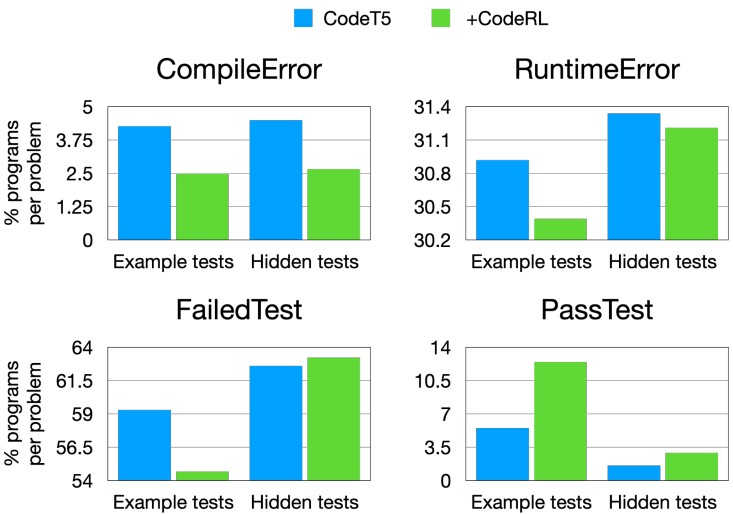

Figure 5: **Qualitative results of CodeT5 and CodeT5+CodeRL:** We generate 200 programs per test sample and report the % programs per sample by their unit test signals, including CompileError, RuntimeError, FailedTest, and PassedTest.

tests and hence, applying our CS procedure might lead to false positive samples for regeneration. We recommend additional methods to improve example unit tests, such as through data augmentation by mutating input/output pairs [Austin et al., 2021].

### D.6 Example Generated Programs and Qualitative Analysis

Figure 6 to 8 show examples of programming problems from the APPS benchmark and corresponding programs generated by CodeT5 variants. Specifically, based on the same foundation pretrained CodeT5 (pretrained with GCPY data and NTP objective), we compare the CodeT5 model that is finetuned by $\mathcal{L}_{ce}$ only and another model that follows our CodeRL framework. In CodeRL+CodeT5, we further show programs before and after applying the CS procedure. The example programs show that applying CodeRL can improve the quality of generated programs and the CS procedure further improves the functional correctness of the programs. For instance, in Figure 8, CodeT5 model misunderstands the problem and focuses on finding the greatest common divisor between $a$ and $b$ only. Instead, the CodeRL model avoids this mistake and tackles the problem based on the *factorials* of $a$ and $b$. In Figure 7, we note that the CS procedure improves the program by reordering the `elif` code blocks. The resulting program is more correct and is able to pass all hidden unit tests.

We also found that CodeRL can improve the efficiency of the programs, an important quality in complex programming problems. For instance, in the interview-level programs in Figure 8, we note that without applying CS, the generated program is functionally correct but fails during execution due to a timeout error. Applying the CS procedure can condition models on parts of the prior program and (re)generates new tokens to produce a more efficient program. Hence, the resulting final program is able to pass all hidden unit tests (including tests with extremely large values) without timeout errors.

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

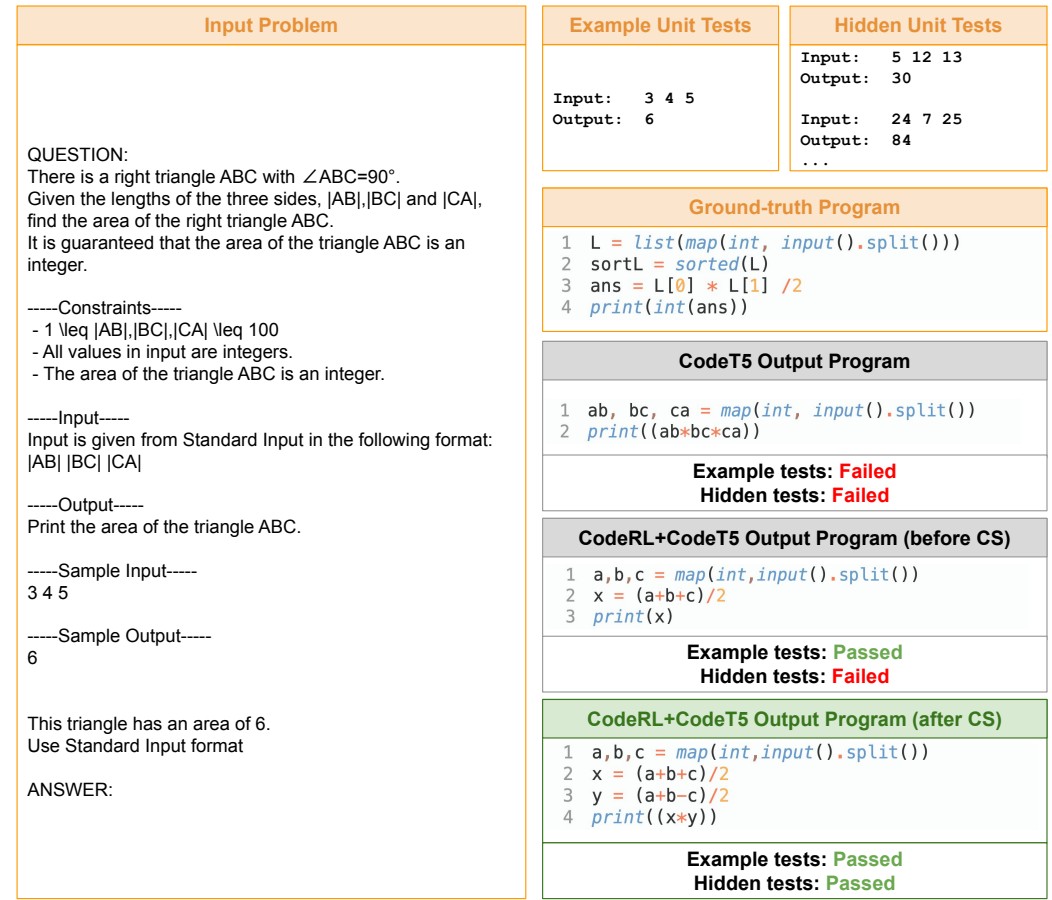

Figure 6: **An example synthesis task from the APPS benchmark and corresponding programs generated by CodeT5 variants:** Without the CS generation procedure, CodeRL+CodeT5 model can generate programs that pass all example tests but fail hidden tests. With the CS generation procedure, the model can condition on prior programs and generate a better program that passes all hidden tests.

D. Hendrycks, S. Basart, S. Kadavath, M. Mazeika, A. Arora, E. Guo, C. Burns, S. Puranik, H. He, D. Song, and J. Steinhardt. Measuring coding challenge competence with apps. *NeurIPS*, 2021.

H. Husain, H. Wu, T. Gazit, M. Allamanis, and M. Brockschmidt. Codesearchnet challenge: Evaluating the state of semantic code search. *CoRR*, abs/1909.09436, 2019.

I. Loshchilov and F. Hutter. Decoupled weight decay regularization. In *ICLR (Poster)*. OpenReview.net, 2019.

S. Lu, D. Guo, S. Ren, J. Huang, A. Svyatkovskiy, A. Blanco, C. B. Clement, D. Drain, D. Jiang, D. Tang, G. Li, L. Zhou, L. Shou, L. Zhou, M. Tufano, M. Gong, M. Zhou, N. Duan, N. Sundaresan, S. K. Deng, S. Fu, and S. Liu. Codexglue: A machine learning benchmark dataset for code understanding and generation. In *NeurIPS Datasets and Benchmarks*, 2021.

K. Papineni, S. Roukos, T. Ward, and W.-J. Zhu. Bleu: a method for automatic evaluation of machine translation. In *Proceedings of the 40th annual meeting on association for computational linguistics*, pages 311–318. Association for Computational Linguistics, 2002.

S. Ren, D. Guo, S. Lu, L. Zhou, S. Liu, D. Tang, N. Sundaresan, M. Zhou, A. Blanco, and S. Ma. Codebleu: a method for automatic evaluation of code synthesis. *arXiv preprint arXiv:2009.10297*, 2020.

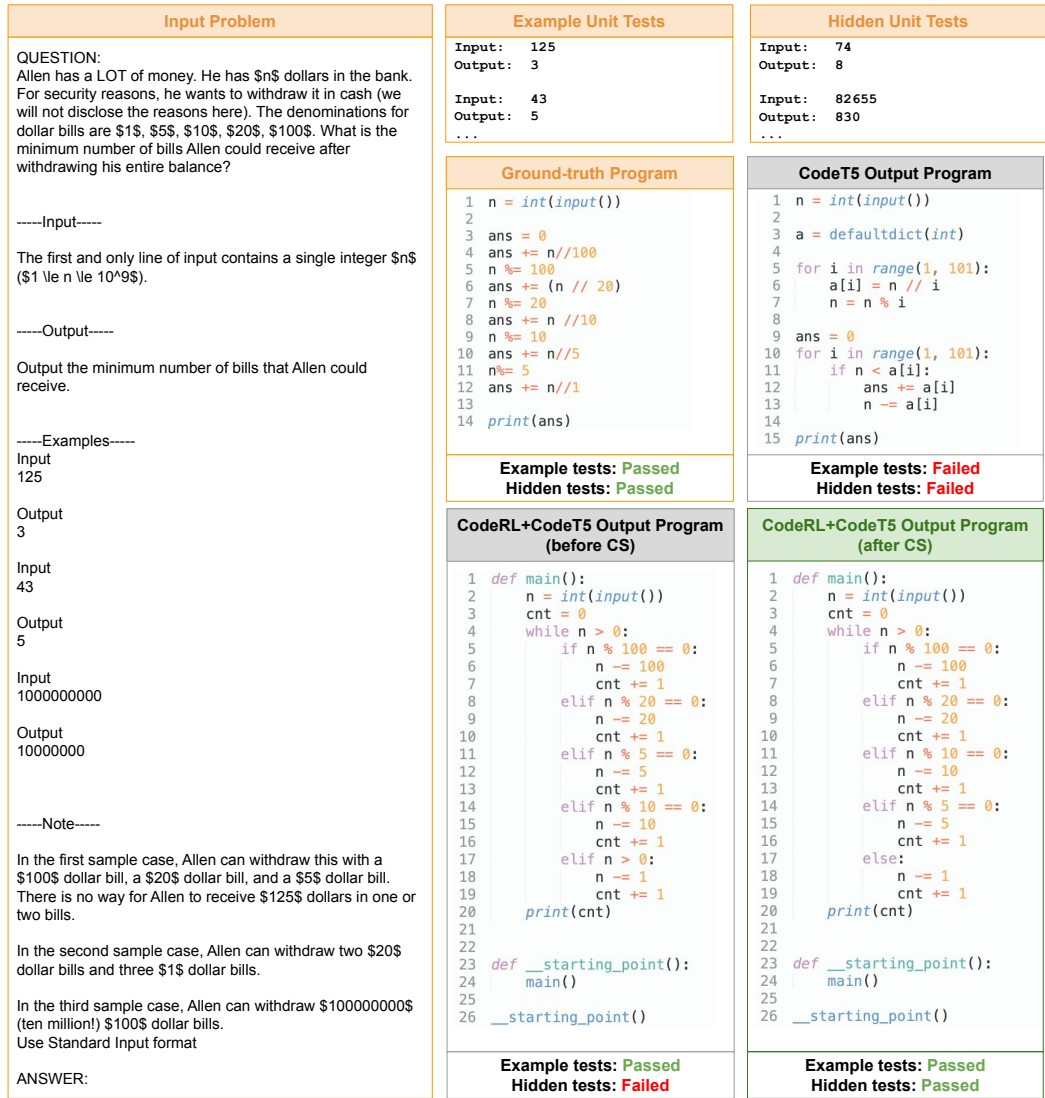

Figure 7: **An example synthesis task from the APPS benchmark and corresponding programs generated by CodeT5 variants:** Without the CS generation procedure, CodeRL+CodeT5 model can generate programs that pass all example tests but fail hidden tests, especially those of corner cases. With the CS generation procedure, the model can condition on prior programs and refine the code. Specifically, we observe the model can simply reorder the `elif` blocks between line 11 and 15 to fix the error. The resulting program is functionally correct and passes all hidden tests.

199  S. J. Rennie, E. Marcheret, Y. Mroueh, J. Ross, and V. Goel. Self-critical sequence training for image
200      captioning. In *Proceedings of the IEEE conference on computer vision and pattern recognition*,
201      pages 7008–7024, 2017.

202  B. Wang and A. Komatsuzaki. GPT-J-6B: A 6 Billion Parameter Autoregressive Language Model.
203      `https://github.com/kingoflolz/mesh-transformer-jax`, May 2021.

204  X. Wang, W. Chen, J. Wu, Y.-F. Wang, and W. Y. Wang. Video captioning via hierarchical rein-
205      forcement learning. In *Proceedings of the IEEE Conference on Computer Vision and Pattern*
206      *Recognition*, pages 4213–4222, 2018.

207  Y. Wang, W. Wang, S. R. Joty, and S. C. H. Hoi. Codet5: Identifier-aware unified pre-trained
208      encoder-decoder models for code understanding and generation. In *EMNLP (1)*, pages 8696–8708.

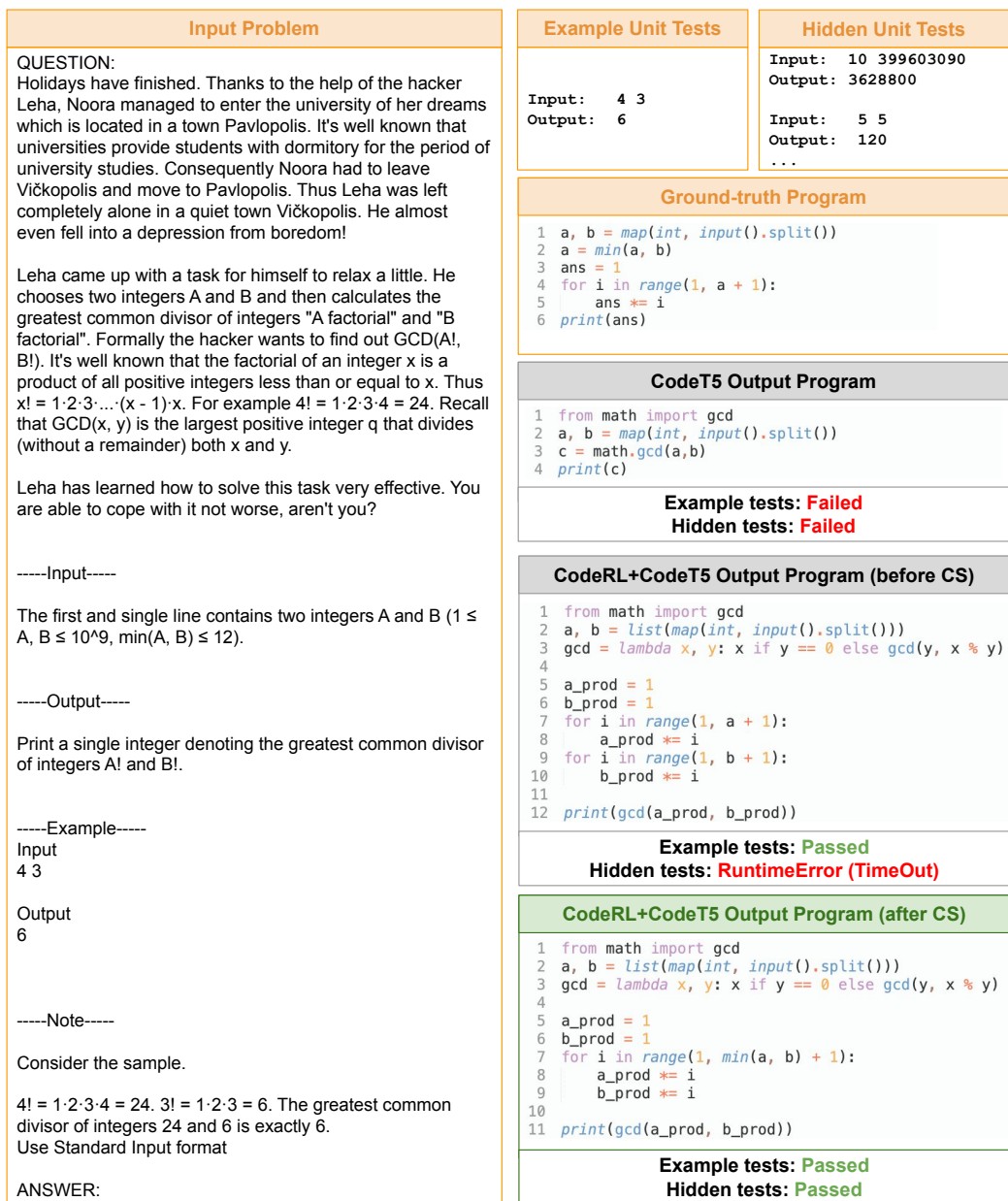

Figure 8: **An example interview-level APPS programming task and programs generated by CodeT5 variants:** The program generated by CodeT5 model fails all unit tests while CodeRL+CodeT5 (without CS generation) can generate a functionally correct program. However, this program leads to runtime errors in extreme test cases. After applying the CS generation procedure, the program is improved and able to pass all hidden unit tests.

209    Association for Computational Linguistics, 2021.