# OpenReview forum: "CodeRL: Mastering Code Generation through Pretrained Models and Deep Reinforcement Learning"
_NeurIPS.cc/2022/Conference — NeurIPS 2022 Accept_

### Official Review · Reviewer_5RxT · 2022-07-09

**Rating:** 7
**Confidence:** 4
**Soundness:** 3 good
**Presentation:** 3 good
**Contribution:** 3 good

**Summary:**

This paper addresses the task of synthesizing programs from specifications consisting of natural language and unit tests. It combines unsupervised pre-training on large code datasets crawled from GitHub, supervised fine-tuning (imitation learning) on a dataset of natural language specifications and ground-truth programs, then reinforcement learning using the unit tests from the same dataset. The authors evaluate on the APPS dataset using various baselines and report state-of-the-art results.

**Questions:**

- Why was the critic trained using max pooling over the outputs of all tokens and then used by taking each token's output individually (before pooling)? Aren't there more principled methods to estimate the intermediate returns (i.e. the value function)?
- How many rounds were done for the critic-guided resampling? Which samples are truncated, dropped entirely, or kept unmodified for the final set?
- If the examples in the training set are too hard for the initial model trained with imitation learning (e.g. almost no samples from the initial model pass all unit tests), the reward will always be negative and the reinforcement learning will not work. Can we quantify how much the distribution of different difficulties in the training set helped? For example, what would happen if the model was only trained on competition-level problems?
- It may not always be convenient to have unit tests available as part of the specification. How well would the critic sampling approach work without unit tests for the test-time examples and only using the information from the critic, and how might that affect the results on the pass@k and 1@k metrics?

**Limitations:**

I thought the description of societal/broader impacts was sufficient.

I didn't find a description of how long the policy gradient was run and how much computation was used for the RL step. There is also no information on how different random seeds may affect the fine-tuning or reinforcement learning part (I understand that it would be too expensive to repeat the pre-training multiple times). I hope the authors can provide such information in future revisions of the paper.

**Strengths And Weaknesses:**

# Originality
Using reinforcement learning for program synthesis is not novel in itself, but as the paper mentions, past works have used domain-specific and toy languages rather than general-purpose languages like Python. This work also differs from others using RL for sequence generation in that it does not rely on smoother measures of quality like BLEU score to have a shaped reward function.

Most of the techniques used in the paper seem to originate from other work, but I think the authors used them competently to achieve an interesting result.

# Quality
The authors report state-of-the-art results on the APPS dataset, which was released in 2021 and contains coding problems of various difficulty. They also show impressive results on the MBPP benchmark which is also a Python dataset of programming problems like APPS, without specific training for the MBPP benchmark, 63.0% pass@80 accuracy, exceeding a much larger 137B-parameter model's 61.4% (which was the best result at the time of the paper submission; since then, [PaLM-Coder](https://arxiv.org/abs/2204.02311) reports 80-85% pass@k for two other, large models).

One weakness of the work is that they only evaluated using one base, pretrained model (an updated version of CodeT5) and using one fine-tuning dataset (APPS). I think the work would have been stronger if they showed the applicability of the method on more models and on more datasets, but the existing result is enough.

# Clarity
Overall I felt that the paper is clear and well-explained. My main confusion was regarding Critic Sampling, which is one of the main methodologically novel parts of the work. For example, it was not clear how many rounds the sampling approach should run. The mechanism can be explained using pseudocode to increase clarity.

# Significance
The growing scale of transformer models and their pretraining datasets has led to significant increases in their ability to synthesize programs in general-purpose languages, but improvements from larger scale alone are not sustainable. Works like this suggest other directions to improve performance further.

Unfortunately, it was not entirely clear to me from the ablation studies about how much of the improvement should be credited to the use of RL versus the use of critic sampling. The ablation studies on RL versus critic sampling are done on different metrics (pass@1, pass@5 versus pass@200, pass@1000, 1@1000). The significance of the work would be clearer otherwise.

---

> ### Author Response · Authors · 2022-08-02
> **Response to Reviewer-5RxT**
>
> **Q1: “My main confusion was regarding Critic Sampling, which is one of the main methodologically novel parts of the work.”**
>
> Our critic sampling is essentially a new round of code generation in which the language model conditions new tokens on programs selected from the prior generation round. These selected programs are chosen based on their success example unit test outcomes (i.e. all failed samples are removed), and then truncated by critic scoring to create sub-programs as seed sequences. We can consider this new generation round as a program completion task in which parts of the programs are provided from potentially correct samples, and models are required to fill in the remaining code.
>
> Many rounds of generations can be run to keep improving the output programs. However, in practice, we found that one additional generation round is sufficient to obtain good quality new programs. Note that in the first round of generation, if no samples pass example unit tests, we do not apply critic sampling and simply use all available programs for evaluation.
>
> **Q2: “It was not entirely clear to me from the ablation studies about how much of the improvement should be credited to the use of RL versus the use of critic sampling.”**
>
> Thank you for your comment. The overall performance gains can actually be attributed to both the RL training objective during training time and the critic sampling procedure during test time. About the ablation studies of critic sampling, in addition to the performance gains as shown in Table 4, the positive impacts of critic sampling on pass@1 and pass@5 are indicated by comparing the results of Table 1 and Table 2 (row D, in which we only used conventional beam search decoding without critic sampling). We can observe that, using critic sampling, model performance increases from 2.2% pass@1 (3.1% pass@5) to 2.57% pass@1 (6.21% pass@5). We will elaborate this comparison clearer by using the additional content page in the final version of the paper.
>
> **Others:**
>
> *“Why was the critic trained using max pooling over the outputs of all tokens and then used by taking each token's output individually (before pooling)?”*
>
> The critic is trained using max pooling as we want to focus on the supervision signals (e.g. unit test outcomes) for whole programs only. Note that we could but did not choose to apply unit tests and obtain supervision signals on partial programs as this approach would heavily penalize many sub-sequences (even though the eventual programs are correct). As a result, many intermediate returns will be biased by low return estimates. In our work, we chose to use max pooling to train critic models, and then use each token’s hidden states to obtain intermediate returns of sub-sequences.
>
> *“If the examples in the training set are too hard for the initial model trained with imitation learning… the reward will always be negative and the reinforcement learning will not work.”*
>
> In practice, we jointly trained the RL loss and the conventional supervised learning loss (next-token prediction using ground-truth solutions). Therefore, for too hard samples with no positive synthetic samples, we can still train models with the normal non-RL loss.
>
> *“How well would the critic sampling approach work without unit tests for the test-time examples and only using the information from the critic?”*
>
> Thanks for your comment. Note that the critic’s task (predicting right or wrong programs) is a nontrivial and very challenging task. Without unit tests, using information from a critic model alone often leads to many false positive filtered samples. Therefore, using unit tests in our critic sampling approach can effectively filter out many wrong samples and make it easier for the critic to select high-quality program samples among the resulting set.
>
> *Additional details of finetuning:*
>
> We run the policy gradient for up to 10 training epochs until both RL loss and NTP loss are converged. In general, we do not find significant impacts (by monitoring training losses) when using different random seeds for finetuning.

---

> > ### Comment · Reviewer_5RxT · 2022-08-07
> > **Reply**
> >
> > Thank you for your response and further explanation about the paper.
> >
> > Thank you for clarifying that there is only one round for critic sampling and for describing how to see the impact of critic sampling within the paper results. To further elucidate critic sampling, I would still recommend some kind of pseudocode description, which can be further supplement the prose description in lines 183-204. You can also place numbers in Figure 3, in the boxes or arrows, to make the sequence of events clearer. In the example generated programs shown in the supplementary material, you can further highlight the edits made by the critic sampling procedure.
> >
> > > In practice, we jointly trained the RL loss and the conventional supervised learning loss (next-token prediction using ground-truth solutions).
> >
> > I see that is mentioned in lines 222-223 of the paper already. I would recommend more details in the paper about the relative time spent on the three phases of training ($L_{ce}$, training the critic, then using $L_{ce}$ together with $L_{rl}$). Currently I understand that the first and third phases can have up to 10 epochs.
> >
> > > Without unit tests, using information from a critic model alone often leads to many false positive filtered samples. Therefore, using unit tests in our critic sampling approach can effectively filter out many wrong samples and make it easier for the critic to select high-quality program samples among the resulting set.
> >
> > By "false positive", do you mean that the critic will often accept programs which are actually incorrect? In any case, I understand that critic sampling always uses example unit tests to produce the final output. How did you separate the unit tests provided with each example in APPS to form the example versus hidden unit tests? In Table 1(a), did the CodeRL+CodeT5 method have additional information available to it (the example unit tests) which the other methods lacked?

---

> > > ### Author Response · Authors · 2022-08-09
> > > **Response by Authors**
> > >
> > >  *Critical sampling procedure:*
> > >
> > > Thank you for your suggestions! Pseudocode is a great idea! We will create and add one to the final version of the paper using the extra content page. We will also incorporate other presentation ideas to make the paper clearer.
> > >
> > > *Training details:*
> > >
> > > For the second phase (i.e. training the critic), we train the critic model (either a GPT2-small or a CodeT5-base) for up to 10 epochs. We found that in this phase, the critic model usually converges quite earlier than the 10-epoch training limit. We added some of these training time details into the current Supplementary Material (Appendix C, L71-91).
> > >
> > > *By "false positive", do you mean that the critic will often accept programs which are actually incorrect?*
> > >
> > > Yes, it can lead to either false positive or false negative filtering. In our approach, we use the example unit tests to filter for positive program samples. Then the critic is used to select sub-sequences among these filter samples as the seeds to condition the LM in the next round of generation. The task of selecting sub-sequences demands less from the critic and allows some room for errors without affecting the performance significantly.
> > >
> > > *How did you separate the unit tests provided with each example in APPS to form the example versus hidden unit tests? In Table 1(a), did the CodeRL+CodeT5 method have additional information available to it (the example unit tests) which the other methods lacked?*
> > >
> > > In each APPS problem, the problem description often contains some example input-output pair. We used Python string-based methods to detect these input-input pairs and extract them to form example unit tests (we will release this preprocessing code together with the paper). In the APPS benchmark, for each problem, the hidden unit tests are already separated from the problem description and provided as a list of input-output pairs. All models in Table 1a are provided with the same information available (the problem description, including embedded input-output pairs). For all models, we follow the benchmark's default way to construct the input sequence, which is the problem description including any example input-output pairs.

---

### Official Review · Reviewer_Z533 · 2022-07-11

**Rating:** 7
**Confidence:** 4
**Soundness:** 3 good
**Presentation:** 3 good
**Contribution:** 4 excellent

**Summary:**

This work builds an actor-critic RL system on top of a pretrained language model for code. The critic is finetuned from unit test execution results of programs sampled from the language model. It's trained to estimate eventual outcome (execution correctness or any errors) at each location of the partial sampled program. The resulting system, when finetuned on the competition dataset APPS, outperforms all known finetuned and few-shot baselines of different sizes. Its performance also transfers zero-shot to a different benchmark MBPP, although with a high degree of potential overlap with finetuning data.

**Questions:**

1. Please address the MBPP/APPS overlap described above through a different lens: for a given pass@k metric, among the correct programs produced by CodeRL, which fraction of lines occurs in the training set of APPS?
2. What's the impact of critic sampling on pass@1 and pass@5 metrics?
3. Does the critic learn to correctly predict erroneous locations in wrong programs? Its max-pool mechanism and ability to predict outcomes for partial programs suggests a possibility to interpret the critic as an error localization model. Can you plot how the critic's predictions change over the course of the program's length, and when do they start matching the ground truth?
4. How well does a critic trained on a single model's predictions transfer to other models?

**Strengths And Weaknesses:**

Strength:
- Great idea and execution. The novelty of actor-critic setup is limited but its application to execution-guided code generation for general-purpose programming languages is, as far as I know, the first of its kind.
- Well-conducted execution with multiple metrics and comparable baselines.
- Currently the best system for competitive code generation for both precision (n@k) and recall (pass@k) metrics, at smaller model scale than LM-only competitors.

Weaknesses:
- The effect of scaling is not studied as comprehensively as for baseline language-model only systems. The only two actors evaluated are CodeT5 @ 770M and GPT-J @ 6B, and it's unclear whether the signal learned by CodeRL critic transfers across models and/or model scales. Since training CodeRL is rather resource-intensive, demonstrating that a single critic model could be trained and applied to other generative LMs with potentially different error profiles would be valuable.
- The "Critic Sampling" procedure is rather ad-hoc, and adds a substantial amount of computational overhead for a limited benefit. Its impact is only demonstrated at larger sample sizes (200 or 1000 in Table 4) and adds up to 1.5 points. No ablation data is given for it for the main metrics of Table 1.
- The transfer results of APPS to zero-shot MBPP show an interesting learning (namely, that learning from competition problems of APPS can transfer to an unrelated dataset without finetuning) but the overlap results in Appendix C.2 question this learning.
Language models are robust to syntactic differences (e.g., internal whitespace, parentheses, semantically equivalent rewrites, and variable names). Even ignoring this, the only-whitespace normalized overlap in Figure 2 of C.2 has 12% of MBPP with more than half of their lines appearing in training data of APPS.
Hypothetically, if (a) these programs were spread uniformly across MBPP problems, and (b) these 12% problems were the ones newly solved by CodeRL, that would account entirely for the difference between "Raw CodeT5 finetuned" and "CodeT5+CodeRL zero-shot" in Table 4. In practice, these hypotheticals are likely not as severe, but they must both be measured in order to properly appreciate the MBPP transfer result.

---

> ### Author Response · Authors · 2022-08-02
> **Response to Reviewer-Z533**
>
> **Q1: The effect of scaling is not studied as comprehensively as for baseline language-model only systems…demonstrating that a single critic model could be trained and applied to other generative LMs with potentially different error profiles would be valuable.”**
>
> Thanks for your comment. In Table 3, we conducted experiments of CodeRL integrated with GPT-Neo. In the Supplementary Material, Appendix D.3 and Figure 3, we tried to integrate CodeRL with GPT-J and analyzed the model performance on competition-level programming tasks. Both GPT-Neo (2.7B) and GPT-J (6B) are larger than our main CodeT5 (770M). We will consider more experiments to study the effect of scaling, e.g. through a single critic model.
>
>
> **Q2: “The "Critic Sampling" procedure is rather ad-hoc, and adds a substantial amount of computational overhead for a limited benefit.”**
>
> Together with improvement on model training, we introduced Critic Sampling to systematically utilize unit test signals and critic models during inference as well. This strategy positions CodeRL as a general dual framework of model training and testing for code generation, applying to foundation models like CodeT5. In addition to the performance gains as shown in Table 4, the positive impacts of critic sampling on pass@1 and pass@5 are indicated by comparing the results of Table 1 and Table 2 (row D, in which we only used conventional beam search decoding without critic sampling). We can observe that, using critic sampling, model performance increases from 2.2% pass@1 (3.1% pass@5) to 2.57% pass@1 (6.21% pass@5). We will elaborate this comparison clearer by using the additional content page in the final version of the paper.
>
> **Q3: “The transfer results of APPS to zero-shot MBPP show an interesting learning…but the overlap results in Appendix C.2 question this learning.”**
>
> Although 12.6% of MBPP programs have more than half of the lines that can be found in the APPS training set, this is computed based on a loose requirement where lines do not have to be consecutive. If we require them to be consecutive, there is no overlap of more than two lines.
>
> We counted their occurrences in the training and test set and found that they have a reasonably balanced distribution, i.e., 49/374=13.1% in the training set and 71/500=14.2% in the test set.
>
> We analyzed the 108 problems that CodeRL can correctly predict while the finetuned CodeT5 fails to do so, where we observed only 21 of them (19.4%) fall into the above 12.6% subset.
>
> **Others:**
>
> *“Does the critic learn to correctly predict erroneous locations in wrong programs?”*
>
> We train critics only to predict the overall test outcomes of the whole program. We did not train the critic to localize the error positions in wrong programs as we found it is quite tricky to obtain reliable supervision signals of error positions (by lines or by tokens).

---

### Official Review · Reviewer_Ukwi · 2022-07-12

**Rating:** 7
**Confidence:** 4
**Soundness:** 3 good
**Presentation:** 3 good
**Contribution:** 3 good

**Summary:**

This work uses an existing pre-trained language model as an actor that synthesizes code to train a critic based on the results of unit tests.
The actor generates samples and the critic evaluates if unit tests are correct. The paper is well-written.

**Questions:**

1. Why not use unit tests as examples in the prompt or in few-shot learning?
2. How sensitive is the work to the reward values?
3. Which Codex version is used and with what hyperparameters?
4. Why not compare the same actor-critic formulation using different language models?


**Limitations:**

The authors have adequately addressed the limitations.

**Strengths And Weaknesses:**

Strengths:
1. Using unit tests and their results as a reward signal in this fashion is a first.
2. Working with pre-trained models, without retraining, makes the approach independent of the model and usable.
3. The results present an efficient generation process, with an order magnitude fewer programs generated, compared with AlphaCode, 1000 generations instead of 50,000 for reaching a similar result.
4. The experiments reduce compiler errors, push the errors down the pipeline, and increase runtime errors, and breakdown of errors.

Weaknesses:
1. Missing comparison with using unit tests in few-shot learning.
2. Missing comparison with CodeRL component on top of other baselines.
3. Missing comparison with of state-of-the-art model baseline: Incoder, CodeGen, and GPT-NeoX.
4. No results on Codeforces data, though discussed in the introduction.
5. Only Python errors are reported in the results, missing errors in other languages.

---

> ### Author Response · Authors · 2022-08-02
> **Response to Reviewer-Ukwi**
>
> **Q1: “Missing comparison with using unit tests in few-shot learning.”**
>
> A problem description usually embeds example input and output pairs (i.e. example unit tests). Therefore, by default, we already included unit tests in the prompt in the input sequence. Few-shot learning is not a focus of our paper as we investigate an RL-based finetuning approach. Nevertheless, we would like to extend our work in few-shot setups in the future, especially on very large language models.
>
> **Q2: “Missing comparison with CodeRL component on top of other baselines.”**
>
> In Table 3, we conducted experiments of CodeRL integrated with GPT-Neo. In the Supplementary Material, Appendix D.3 and Figure 3, we tried to integrate CodeRL with GPT-J and analyzed the model performance on competition-level programming tasks.
>
> **Q3: “Missing comparison with state-of-the-art model baseline: Incoder, CodeGen, and GPT-NeoX.”**
>
> Thank you for your comment. We are running some experiments on these newly released baselines and will add more results in the future.
>
> **Q4: “No results on Codeforces data, though discussed in the introduction.”**
>
> Codeforces is one of the major data sources of APPS.
>
> **Q5: “Only Python errors are reported in the results, missing errors in other languages.”**
>
> We will explore program synthesis in other languages. One of the bottlenecks has been the lack of released evaluation tools to run programs against unit tests for other languages (except Python).
>
> **Others:**
>
> *"How sensitive is the work to the reward values?"*
>
> We chose reward values such that they range from -1 to 1 for all possible unit test outcomes. We experimented with heavier penalties e.g. -3 for compile errors, -2 for runtime errors, and -1 for failed tests. However, we found such a penalty-focused reward scheme often leads to a very unstable finetuning process.
>
> *"Which Codex version is used and with what hyperparameters?"*
>
> We directly reported the results of Codex on APPS as reported by the original authors. Please refer to the Codex paper for more details of model versions and hyperparameters.

---

### Official Review · Reviewer_paYh · 2022-07-12

**Rating:** 4
**Confidence:** 4
**Soundness:** 3 good
**Presentation:** 4 excellent
**Contribution:** 2 fair

**Summary:**

This paper addresses the problem of neural program synthesis, whose aim is to train a neural network to synthesize programs from problem specifications. To this end, the paper proposes a framework that leverages large-scale pre-trained language models and directly optimizes the functional correctness of generated programs using reinforcement learning. The experiments show that the propsoed framework outperforms baselines on the APPS benchmark and demonstrates reasonable generalization on the MBPP benchmark. Ablation studies justify the effectiveness of the design choices and the proposed techniques. I believe this work explores a promising research direction and proposes a reasonable framework to address the problem. Yet, I am concerned with insufficient novelty and seemingly inconclusive experimental results.

**Questions:**

Described in the Strengths And Weaknesses section.


**Limitations:**

Described in the Strengths And Weaknesses section.


**Strengths And Weaknesses:**

## Paper strengths and contributions
**Motivation and technical contribution**
- The motivation for leveraging pre-trained large language models and RL to improve program synthesis performance is convincing and this paper proposes a reasonable framework to implement this idea.

**Clarity**
The overall writing is clear. The authors utilize figures well to illustrate the ideas.

**Ablation study**
- Ablation studies are comprehensive. The proposed framework consists of multiple components and the provided ablation studies help analyze the effectiveness of each of them, including the cross entropy loss and the RL loss, the design of return estimate, the choice of pre-training approaches, and the critic sampling.

**Experimental results**
- The presentation of the experimental results is clear.
- The experimental results show that the proposed framework outperforms the baselines in terms of both $pass@k$ and $n@k$ on APPS.

**Reproducibility**
Given the clear description in the main paper and the details provided in the appendix, I believe reproducing the results is possible.

## Paper weaknesses and questions

**Citation format**
The citation format does seem right. Citations in NeurIPS papers are referred to by numbers, not the first authors' last name.

**Novelty & contribution**
Overall, I do not find enough novelty in any aspect while the overall effort of this paper is appreciated. Specifically,
- Using RL for improving program synthesizers' performance has been studied in "Bridging reinforcement learning and maximum marginal likelihood", "Leveraging grammar and reinforcement learning for neural program synthesis", etc. Adapting this to generating Python code does not seem to be particularly novel.
- Leveraging unit tests and the results to filter and cluster generated programs has been proposed in "Competition-level code generation with AlphaCode".

**Related work**
The descriptions of the related work are not comprehensive. Several important prior works in neural program synthesis are omitted from the paper, including "Neural Scene De-rendering", "Neural Program Synthesis from Diverse Demonstration Videos", "Learning to Describe Scenes with Programs", "Learning to Infer and Execute 3D Shape Programs", etc.

**Two critics**
The paper mentions that two separate critics are used. I am curious about the intuition behind it and am interested to know whether this contributes to the performance gain.

**CodeContests dataset**
It would be more informative if the authors can also evaluate their proposed framework on the CodeContests dataset, which can be found on [Github](https://github.com/deepmind/code_contests). It would be easier to compare the proposed framework to AlphaCode.

**Low accuracy**
The reported accuracy is rather low (mostly < 10\%) for all the methods. It would make sense to increase $k$ of $pass @ k$ and increase both $n$ and $k$ of $n @ k$.

**Standard deviation of accuracy**
The reported accuracy is rather low for all the methods. In this case, I believe showing the standard deviation of accuracy would be important to compare different methods.

**Comparison to AlphaCode**
AlphaCode paper reports the need for millions of program samples to achieve satifactory performance. It would make sense to increase $k$ of $pass @ k$ and see if the conclusion holds.

---

> ### Author Response · Authors · 2022-08-02
> **Response to Reviewer-paYh (1/n)**
>
> **Q1: “Using RL for improving program synthesizers' performance has been studied…Adapting this to generating Python code does not seem to be particularly novel.”**
>
> Thank you for your comment. Compared to the prior related work, our work is different in 3 main points:
> - Methodology: prior work such as [1, 2] was not used together with pretrained language models for code generation. Existing language models have achieved notable results but they do not fully use important signals such as unit test outcomes to improve the quality of programs. By focusing our RL framework on language models such as CodeT5, we can leverage the powerful capabilities of these models learned on huge amounts of data, while improving their performance to generate high-quality programs.
>
> - Definition of program synthesis task: compared to related work such as [2], our work focuses on program synthesis with problem specification in natural language rather than input-output (IO) pairs. Applying prior models to tackle problems described in natural language is not trivial, as they would fail to interpret the problems, especially competition-programming problems with complex natural specifications.
>
> - Complexity of program synthesis task: prior papers such as [1, 2] are limited to domain-specific and toy languages which are very different from programming languages like Python. Generating code in high-level and general-purpose programming languages involves a very large search space of possible programs. Applying prior RL-based approaches to generate such programs is not trivial as these models are originally designed with domain-specific language features, e.g. program path exploration and grammar checks, that are not readily accessible in languages like Python. Finally, in our paper, we focus on Python programming tasks, and specifically on highly complex problems e.g. those from programming competitions on Codeforces.
>
> Overall, as recognized by Reviewer-Z533, our CodeRL is the first application of the RL framework to execution-guided code generation for general-purpose programming languages. CodeRL is not just a holistic approach to program synthesis through combining RL with pretrained language models, but also introduces a novel way to leverage the unit testing signals in both learning and inference stages.
>
> *[1] “From Language to Programs: Bridging Reinforcement Learning and Maximum Marginal Likelihood”*
>
> *[2] “Leveraging grammar and reinforcement learning for neural program synthesis”*
>
> **Q2: “Leveraging unit tests and the results to filter and cluster generated programs has been proposed in "Competition-level code generation with AlphaCode".”**
>
> While AlphaCode introduced unit testing mainly as a post-processing step to cluster and filter generated programs, our work utilizes unit tests as part of the model generation procedure where the model is allowed to regenerate and improve output programs. Specifically, we propose to use the prior filtered programs as seeds to initialize decoding tokens and let the model regenerate the remaining tokens. Secondly, compared to AlphaCode, our approach does not require generating a large number of additional unit tests, which is essential to obtain diverse clusters for sampling in AlphaCode. Note that generating unit tests is not trivial, especially for competition-level programming tasks.
>
> **Q3: “It would be more informative if the authors can also evaluate their proposed framework on the CodeContests dataset, which can be found on Github.”**
>
> We did try to evaluate on the CodeContests but faced a major obstacle of the incomplete released evaluation scripts for this dataset. To be more specific, results in the AlphaCode paper are reported based on the model trained on ground-truth programs in three PLs (Python, C++, and Java), however, only the evaluation tool for Python has been released. The authors mentioned the difficulties of releasing their C++ sandbox evaluation tool (“significantly more complicated, and more coupled with internal tools”) in a GitHub issue thread. Note that C++ takes up the largest proportion of the training data. As such, it is not trivial to make a fair comparison with AlphaCode on this dataset and we did not notice any follow-up work including the CodeContests evaluation. By contrast, APPS is a more widely adopted benchmark by works such as Codex,  AlphaCode, and more recently [3]. In addition, APPS has a much more comprehensive testing set compared to CodeContests (5000 vs. 165 test instances).
>
> *[3] “Fault-Aware Neural Code Rankers”*

---

> > ### Author Response · Authors · 2022-08-02
> > **Response to Reviewer-paYh (2/n)**
> >
> > **Q4: “AlphaCode paper reports the need for millions of program samples to achieve satisfactory performance. It would make sense to increase k of pass@k and see if the conclusion holds.”**
> >
> > Indeed, increasing the generation budget (in millions of programs) can push the results to a more satisfactory level. However, we also want to consider whether pushing the generation budget k is the right research direction and whether this budget is viable. For a much larger test set in APPS (of 5000 samples vs. 165 samples in CodeContests where AlphaCode reported results for one million program samples), we expect scaling up the generation to this amount would take too much computation resource on APPS. Moreover, pushing k to million scales is not very practical and unlikely to be affordable for many users in the real world.
> >
> > **Q5: “The reported accuracy is rather low (mostly < 10%) for all the methods. It would make sense to increase k of pass@k and increase both n and k of n@k.”**
> >
> > Thanks for your comment. We will attempt to increase the generation budget k and n and report the results. However, we also want to underline the viability of generation budget k in real-world scenarios and focus on experiments at more practical and affordable levels. In the current results, our best model performance is 19.36% of pass@1000 and 11.61% of 5@1000. We added additional results of baseline models of pass@1000 in Table 1 and still observed the significant performance gains of our approach.
> >
> > **Q6: “I believe showing the standard deviation of accuracy would be important to compare different methods.”**
> >
> > We follow the prior approaches which did not report the standard deviation of accuracy. One reason is due to the computation cost during inference time to generate a large number of program samples through pretrained language models. Also note that by default, the pass@k metric is already normalized by the total number of sets of k programs among the generated program population.
> >
> > **Others:**
> >
> > *“The paper mentions that two separate critics are used. I am curious about the intuition behind it and am interested to know whether this contributes to the performance gain.”*
> >
> > During training time, we want a critic that can predict four different types of unit test outcomes, namely compile errors, runtime errors, failed tests, and passed tests, to obtain training signals (return estimates) in fine-grained details. During test time, we only care whether a program is functionally correct or not and hence, chose to have another critic for a binary prediction task. In practice, we do not see too significant performance differences when using the same critic during both training and inference.
> >
> > *Additional related work:*
> >
> > Thank you for your suggestion. We added the mentioned papers and other related work in the Supplementary Material (due to the current page limit). We will incorporate them in the additional content page in the final version of the paper.

---

### Author Response · Authors · 2022-08-02
**Response to all reviewers**

We thank all the reviewers for their insightful feedback. We are glad that the reviewers recognized the motivation and technical contributions of our work, demonstrated by comprehensive and solid experimental results. We are encouraged that Reviewer-5RxT further recognized the significance of our CodeRL approach to improving code generation systems rather than purely focusing on increasing the scale of models. We address the questions and concerns of each reviewer below.

---

### Meta-Review · Area_Chair_mecu · 2022-08-25

**Recommendation:** Accept
**Confidence:** Certain

**Metareview:**

All reviewers appreciated the overall idea of using reinforcement learning to generate code from language models leveraging the unit tests and critic scores, especially for real-world languages as opposed to most prior work on using RL for program synthesis that generate code in domain-specific languages. The evaluation with extensive ablations was also greatly appreciated. The reviews have lots of suggestions for improving the presentation of the paper and also additional comparisons, which hopefully the authors can incorporate to improve the paper even further.

**Award:**

No

---

### Decision · Program_Chairs · 2022-09-14

Accept